# On Measuring Long-Range Interactions in Graph Neural Networks

**Jacob Bamberger** [* 1]   **Benjamin Gutteridge** [* 1]   **Scott le Roux** [* 1]   **Michael Bronstein** [1 2]   **Xiaowen Dong** [1]

## Abstract

Long-range graph tasks — those dependent on interactions between 'distant' nodes — are an open problem in graph neural network research. Real-world benchmark tasks, especially the Long Range Graph Benchmark, have become popular for validating the long-range capability of proposed architectures. However, this is an empirical approach that lacks both robustness and theoretical underpinning; a more principled characterization of the long-range problem is required. To bridge this gap, we formalize long-range interactions in graph tasks, introduce a **range measure** for operators on graphs, and validate it with synthetic experiments. We then leverage our measure to examine commonly used tasks and architectures, and discuss to what extent they are, in fact, long-range. We believe our work advances efforts to define and address the long-range problem on graphs, and that our range measure will aid evaluation of new datasets and architectures.

## 1. Introduction

Graphs have emerged as a rich data modality, capable of modeling pairwise relationships between entities in diverse applications. Graph neural networks (GNNs) have become the dominant paradigm for learning from such data, with message passing neural networks (MPNNs) (Gilmer et al., 2020) standing out as the most widely used framework. MPNNs operate by iteratively updating a node's representation based on information aggregated from its local neighborhood. This local message-passing mechanism has been highly effective in a wide range of tasks, such as social networks (Monti et al., 2019), computational chemistry (Gilmer et al., 2017) and recommendation systems (Fan et al., 2019).

However, MPNNs struggle to capture **long-range dependencies**, where interactions between distant nodes are critical. These difficulties stem from various well-documented pathologies, such as the difficulty building deep GNNs often attributed to over-smoothing and vanishing gradients (Nt & Maehara, 2019; Oono & Suzuki, 2019; Li et al., 2019), or the inability to propagate information through graph bottlenecks attributed to over-squashing (Alon & Yahav, 2020). These limitations hinder the ability of MPNNs to model tasks that depend on global or long-range interactions.

Various strategies have been proposed to address these issues, but a common theme is the effective *reduction of distance* on a graph — either via explicit static rewiring (Topping et al., 2021; Gasteiger et al., 2019; Barbero et al., 2023), or rewiring of the computational graph via architectural components such as virtual/latent nodes (Gilmer et al., 2017; Southern et al., 2024), fully connected or multi-hop message passing layers (Alon & Yahav, 2020; Abu-El-Haija et al., 2019; Gutteridge et al., 2023), and global attention (Vaswani et al., 2017; Wu et al., 2021; Rampášek et al., 2022). For many of these methods, the argument that they improve performance on long-range tasks is based *solely on empirical performance* on synthetic and real-world benchmarks.

However, most synthetic long-range tasks (Bodnar et al., 2021; Rampášek & Wolf, 2021) are simplistic and depend on long-range interactions *only*, such that a simple rewiring converts them into short-range tasks. Few synthetic tasks acknowledge that graph-structured data and graph tasks are, by design, locally biased: solutions should capture long-range interactions, but should usually prioritize local ones.

Real-world benchmarks face similar problems. While some recent works have introduced and motivated long-range tasks more systematically (Liang et al., 2025), the Long Range Graph Benchmark (LRGB) (Dwivedi et al., 2022) remains ubiquitous. LRGB establishes task long-rangedness via graph size/diameter and domain-specific intuition. These are necessary, but not *sufficient* conditions for qualifying a task as long-range. Furthermore, LRGB's sensitivity to hyperparameter tuning has recently come under scrutiny (Tönshoff et al., 2023), raising doubts about the extent to which its tasks are truly long-range. These issues stem from a lack of theoretical characterization of the long-range issue, which hampers efforts in measuring and addressing it.

---

[*]Equal contribution [1]University of Oxford [2]AITHYRA. Correspondence to: Jacob Bamberger <jacob.bamberger@cs.ox.ac.uk>, Benjamin Gutteridge <beng@robots.ox.ac.uk>, Scott le Roux <sleroux@robots.ox.ac.uk>.

*Proceedings of the 42nd International Conference on Machine Learning*, Vancouver, Canada. PMLR 267, 2025. Copyright 2025 by the author(s).

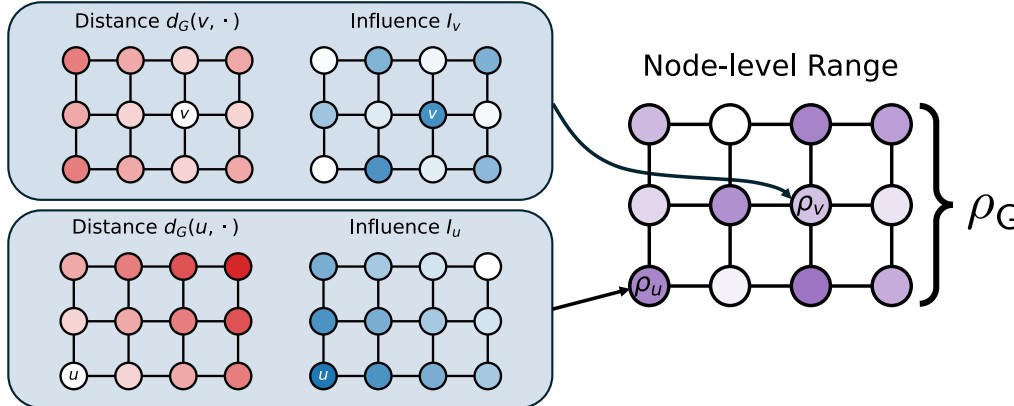

*Figure 1.* Illustration of the range measure on an example grid graph. The distance (left) and influence (center) relative to nodes $v$ (top) and $u$ (bottom) are indicated by node opacity. Taking the node-wise product of distance and influence and then aggregating over the entire graph yields the **node-level range** (right; also represented by opacity), e.g. $\rho_u, \rho_v$. All node ranges can then be aggregated to obtain a **graph range** $\rho_G$. See Table 1 for details on the range measure at different granularities. We use different influence distributions for $u$ and $v$ to show that influence is task-dependent.

In this work, we address these gaps by formalizing, for the first time, the notion of the 'long-range problem' on graphs. Our main contributions are three-fold.

- We introduce a formal definition of long-range interactions in graph tasks, grounded in first principles rather than empirical heuristics. This provides a systematic foundation for analyzing both tasks and architectures.

- We derive a family of *principled, quantitative measures of a GNN's **range*** — the extent to which it captures long-range dependencies. These measures apply to node- and graph-level tasks, support evaluation at multiple granularities — node, graph, and dataset — and function in both inductive and transductive settings.

- We validate our approach through synthetic experiments designed to capture long-range interactions, and then apply it to critically assess the LRGB benchmark.[1]

## 2. Background

**Existing range measures**. A handful of existing works consider the notion of node-to-node interaction more formally. Xu et al. (2018b) introduced the *influence score* as a measure of an output node's sensitivity to an input node as the sum of absolute values of their Jacobian. This approach adapts *influence functions* to graphs, building on the work of Koh & Liang (2017), who originally applied them to interpret neural network predictions. Recent work by Liang et al. (2025) motivates a novel long-range benchmark using a proposed influence-based measure. Our work derives a family of measures from first principles, of which their

---

[1]All code for reproducing experiments is available at https://github.com/BenGutteridge/range-measure

influence-based measure is a specific instantiation. In addition, their work introduces a benchmark, while ours aims at a deeper theoretical investigation of the long-range issue.

Alon & Yahav (2020) introduced a prototypical range measure, the *problem radius*, which, for a task, is defined as the minimal number of hops necessary to solve it. The problem radius is generally unknown and is approximated by tuning the number of layers; larger radii require deeper MPNNs. They also introduce the over-squashing problem, a phenomena that is closely linked to long-range interactions.

Sensitivity analysis by Topping et al. (2021); Di Giovanni et al. (2023a) formalized over-squashing, showing that poorly-connected nodes result in small influence scores. Di Giovanni et al. (2023b) studied pair-wise node interactions by casting model expressivity as a measure of a model's capacity to *mix* node features. Their measure is based on the Hessian of pair-wise node features, focuses specifically on mixing rather than range, and is computed for known tasks; in Section 6.1 we demonstrate how our measure can be applied to models trained on arbitrary tasks.

**Graphs and features**. A graph $G$ is a tuple $(V, E)$ of a set of nodes $V$ and edges $E$. We denote the number of nodes by $n = |V|$, and edges are represented as tuples $(u, v) \in E$. The graphs considered in this work are undirected, and are often represented by an adjacency matrix $\mathbf{A} \in \{0, 1\}^{n \times n}$ with corresponding degree matrix $\mathbf{D} := \mathrm{diag}(\mathbf{A}\mathbf{1})$. We denote the features at node $u$ by $\mathbf{x}_u \in \mathbb{R}^d$ and stack all node features into a matrix $\mathbf{X} \in \mathbb{R}^{n \times d}$. We denote by $\mathcal{N}_k(u)$ the $k$-hop *neighbors* of node $u$, i.e. the nodes at exactly $k$ hops as per shortest-path distance (SPD), and by $\mathcal{N}_{\leq k}(u)$ the $k$-hop *neighborhood* of $u$: all nodes at $k$ *or fewer* hops.

**GNNs**. In many applications, the goal is to perform a prediction task starting from data modeled as a graph $\mathsf{G}$ and node features $\mathbf{X} \in \mathbb{R}^{n \times d}$. For node-level tasks, predictions are made based on a representation for every node $u$ denoted $\mathbf{y}_u \in \mathbb{R}^c$ which can be stacked into a matrix $\mathbf{Y} \in \mathbb{R}^{n \times c}$. GNNs are used to achieve such predictions. Most GNNs, including widely used MPNNs (Kipf & Welling, 2017; Xu et al., 2018a; Bresson & Laurent, 2017) and graph Transformers (GTs) update the nodes features sequentially by computing hidden node features $\mathbf{H}^{(\ell)} \in \mathbb{R}^{n \times d_\ell}$ at each of $L$ layers, with input $\mathbf{H}^{(0)} = \mathbf{X}$ and final layer output $\mathbf{Y} = \mathbf{H}^{(L)}$. For graph-level tasks, a further pooling operation is applied to obtain a single graph-level output $\mathbf{y} \in \mathbb{R}^c$.

# 3. Formalizing the range of a node-level task

To address the challenge of quantifying long-range interactions in GNNs, we propose a measure of the *range* of a task, a GNN, or, more generally, an operator on a graph. This measure captures how strongly distant nodes interact. We derive the measure from first principles as the *unique* measure that satisfies a set of desired properties.

We first propose a definition of the range of a node-level task. By a node-level task on a graph, we mean a map $\mathbf{F}$ transforming an input signal $\mathbf{X} \in \mathbb{R}^{n \times d}$ into an output $\mathbf{Y} = \mathbf{F}(\mathbf{X}) \in \mathbb{R}^{n \times c}$. The range should be applicable to any map $\mathbf{F}$, including linear ones i.e. such that $\mathbf{F}(a\mathbf{X} + b\mathbf{X}') = a\mathbf{F}(\mathbf{X}) + b\mathbf{F}(\mathbf{X}')$ for all $a, b \in \mathbb{R}$ and $\mathbf{X}, \mathbf{X}' \in \mathbb{R}^{n \times d}$. We restrict our attention to linear maps, denoted $\mathbf{L}$, and for simplicity we consider the case $d = c = 1$, where a linear map corresponds to a matrix $\mathbf{L} \in \mathbb{R}^{n \times n}$, and $\mathbf{L}_{uv}$ is the *interaction from $v$ to $u$*. An example of a linear map is the transformation denoted by $\mathbb{1}_{uv}$. This transformation copies the value at position $v$ from the input and places it at position $u$, setting all other values to zero. Specifically, it transforms the matrix $\mathbf{X}$ into a new matrix $\mathbf{Y}$, where $\mathbf{Y}$ is zero everywhere except at index $u$, where $\mathbf{y}_u = \mathbf{x}_v$.

As a task can be long-range around one node and short-range around another, the range measure should be defined *for every node*. Our goal is thus to find a suitable definition of the range of a task $\mathbf{F}$ at a node $u$, denoted by $\rho_u(\mathbf{F}) \in \mathbb{R}_+$.

**Distance metric**. A critical aspect of defining this measure is establishing a notion of *distance* between nodes, with respect to which the range can be defined. While metrics like Euclidean distance are commonly used for sequences and grids, graphs are more complex as there are many choices of metric. Common options include SPD (the length of the shortest path between two nodes in hops), commute-time distance (based on the expected time for a random walk to travel between nodes), and diffusion distances (capturing connectivity in terms of information flow). While SPD is intuitive, recent work suggest that commute-time distance

is better suited to studying how information propagates in MPNNs (Di Giovanni et al., 2023a; Black et al., 2023). For full generality of the measure, we simply assume that we are given a metric $d_\mathsf{G} : \mathsf{V} \times \mathsf{V} \to \mathbb{R}_+$ on the graph, with respect to which we measure the range denoted $\rho_u$.

For our experiments we focus on SPD and resistance distance, denoted by $\rho_u^{\mathrm{spd}}$ and $\rho_u^{\mathrm{res}}$ respectively.

## 3.1. Derivation of node-level range

We first state the desirable properties of the range measure, $\rho_u$, which quantifies the range of interactions received by a node $u$ under a linear task $\mathbf{L}$.

**Property 1 (Locality)** The range at node $u$ should only depend on interactions received by node $u$. i.e. $\rho_u(\mathbb{1}_{wv}) = 0$ if $w \neq u$.
*Motivation:* Ensures that the range measure does not depend on irrelevant parts of the graph.

**Property 2 (Unit interaction)** If the output of node $u$ depends solely on the input of another node $v$ at distance $d_\mathsf{G}(u, v) = r$, then the range should equal $r$. i.e. $\rho_u(\mathbb{1}_{uv}) = d_\mathsf{G}(u, v)$
*Motivation:* Provides a baseline for what constitutes the range of a single interaction.

**Property 3 (Additivity)** The range of disjoint interactions[a] should be the sum of their individual ranges. If $\mathbf{L}^1$ and $\mathbf{L}^2$ are disjoint, then $\rho_u(\mathbf{L}^1 + \mathbf{L}^2) = \rho_u(\mathbf{L}^1) + \rho_u(\mathbf{L}^2)$.
*Motivation:* Ensures that the range appropriately aggregates independent contributions to the output.

**Property 4 (Homogeneity)** Scaling the interaction strength should proportionally scale the range, i.e., $\forall \alpha \in \mathbb{R}, \; \rho_u(\alpha L) = |\alpha| \rho_u(L)$.
*Motivation:* Guarantees that the measure meaningfully reflects the strength of interaction.

---

[a]We say that two linear maps $\mathbf{L}^1$, $\mathbf{L}^2$ consist of *disjoint interactions*, or are *disjoint*, if $|\mathbf{L}_{uv}^1| > 0 \implies \mathbf{L}_{uv}^2 = 0$ and $|\mathbf{L}_{uw}^2| > 0 \implies \mathbf{L}_{uw}^1 = 0$, in other words, they do not capture interactions between the same pairs of nodes.

Remarkably, from these we can uniquely derive $\rho_u$:

**Theorem 3.1.** *Given a graph $\mathsf{G}$, metric $d_\mathsf{G}$, and node $u$, there is a unique range $\rho_u$ defined on one-dimensional linear tasks that satisfies Properties 1–4, and, given a linear task $\mathbf{Y} = \mathbf{L}(\mathbf{X})$, it corresponds to the following:*

$$\rho_u(\mathbf{L}) = \sum_{v \in \mathsf{V}} |\mathbf{L}_{uv}| d_\mathsf{G}(u, v).$$

*Table 1.* Normalized range measures based on the Jacobian and Hessian, for node-level and graph-level tasks, and at node, graph, and dataset granularities. For node-level tasks, $\rho$ denotes ranges derived from the Jacobian (the first-order Taylor series term), which we use as it provides a set of pairwise interactions. For graph-level tasks, the Jacobian does not provide pairwise interaction terms, so we instead use range measures derived from the Hessian (the second-order Taylor term), and denote them by $\eta$.

| | Jacobian-based ($\rho$) | | | Hessian-based ($\eta$) | | |
|---|---|---|---|---|---|---|
| **Granularity** | Node | Graph | Dataset | Node | Graph | Dataset |
| Node-level tasks | $\hat{\rho}_u$ (Eq. 2) | $\hat{\rho}_{\mathsf{G}} := \frac{1}{n}\sum_{u\in\mathsf{V}}\hat{\rho}_u$ | $\hat{\rho}_{\mathcal{G}} := \frac{1}{N}\sum_{i=1}^{N}\hat{\rho}_{\mathsf{G}_i}$ | N/A | N/A | N/A |
| Graph-level tasks | As above; using pre-pooling node features[2] | | | $\hat{\eta}_u$ (Eq. 4) | $\hat{\eta}_{\mathsf{G}} := \frac{1}{n}\sum_{u\in\mathsf{V}}\hat{\eta}_u$ | $\hat{\eta}_{\mathcal{G}} := \frac{1}{N}\sum_{i=1}^{N}\hat{\eta}_{\mathsf{G}_i}$ |

*Proof.* The linear tasks form a vector space which is spanned by the maps $\mathbb{1}_{uv}$. Any linear map $\mathbf{L}$ can therefore be decomposed into a sum $\mathbf{L} = \sum_{ij}\mathbf{L}_{ij}\mathbb{1}_{ij}$. By Property 3, since each $\mathbb{1}_{ij}$ are disjoint, we must have $\rho_u(\mathbf{L}) = \sum_{ij}\rho_u(\mathbf{L}_{ij}\mathbb{1}_{ij})$. By Property 1 the range at $u$ depends only on the terms with $i = u$, i.e. $\rho_u(\mathbf{L}) = \sum_{v\in\mathsf{V}}\rho_u(\mathbf{L}_{uv}\mathbb{1}_{uv})$. By Property 4, we deduce that $\rho_u(\mathbf{L}) = \sum_v |\mathbf{L}_{uv}|\rho_u(\mathbb{1}_{uv})$. By Property 2, we get $\rho_u(\mathbf{L}) = \sum_v |\mathbf{L}_{uv}|d_{\mathsf{G}}(u, v)$. $\square$

**Normalized range**. One consequence of the above properties of $\rho_u$ is that many short-range interactions can increase the range. For example, $m$ disjoint interactions at distance 1 results in a range of $m$, when we might prefer the range to reflect the *average* (or in this case, unanimous) range, which in this case would be 1. As a remedy, we consider a *normalized range*, $\hat{\rho}_u$, by replacing Properties 3 & 4 by:

**Property 5** If $\mathbf{L}^1$ and $\mathbf{L}^2$ are disjoint, then $\forall \alpha \neq 0$, $\beta$, $\rho_u(\alpha\mathbf{L}^1 + \beta\mathbf{L}^2) = \frac{|\alpha|}{|\alpha|+|\beta|}\rho_u(\mathbf{L}^1) + \frac{|\beta|}{|\alpha|+|\beta|}\rho_u(\mathbf{L}^2)$.

From which we also derive a uniqueness result:

**Theorem 3.2.** *Given a graph $\mathsf{G}$, metric $d_{\mathsf{G}}$, and node $u$, there is a unique range $\hat{\rho}_u$, defined on one-dimensional linear tasks, that satisfies Properties 1, 2 & 5, and given a linear task $\mathbf{Y} = \mathbf{L}(\mathbf{X})$ it corresponds to the following:*

$$\hat{\rho}_u(\mathbf{L}) = \frac{1}{\sum_v |\mathbf{L}_{uv}|}\sum_{v\in\mathsf{V}}|\mathbf{L}_{uv}|d_{\mathsf{G}}(u, v).$$

*Proof.* As in the above proof, we decompose $\mathbf{L} = \sum_{ij}\mathbf{L}_{ij}\mathbb{1}_{ij}$. By Properties 3 & 1 & 5 we get $\hat{\rho}_u(\mathbf{L}) = \frac{1}{\sum_{v\in\mathsf{V}}|\mathbf{L}_{uv}|}\sum_{v\in\mathsf{V}}|\mathbf{L}_{uv}|\hat{\rho}_u(\mathbb{1}_{uv})$, and by Property 2 we conclude $\hat{\rho}_u(\mathbf{L}) = \frac{1}{\sum_{v\in\mathsf{V}}|\mathbf{L}_{uv}|}\sum_{v\in\mathsf{V}}|\mathbf{L}_{uv}|d_{\mathsf{G}}(u, v)$. $\square$

**Range of a GNN**. In order to apply the above range to GNNs it suffices to extend this measure to any differentiable map $\mathbf{F}(\mathbf{X}) = \mathbf{Y}$. This is done by applying the above to the Jacobian, namely the best linear approximation of $\mathbf{F}$.

---

[2]For graph-level tasks, the Jacobian-based method is introduced as a computationally more efficient alternative. See Section 6.2.1 for details.

$$\rho_u(\mathbf{F}) := \sum_{v\in\mathsf{V}}\left|\frac{\partial(\mathbf{F}(\mathbf{X}))_u}{\partial\mathbf{x}_v}\right|d_{\mathsf{G}}(u, v), \text{ and} \qquad (1)$$

$$\hat{\rho}_u(\mathbf{F}) := \frac{\rho_u(\mathbf{F})}{\sum_{v\in\mathsf{V}}\left|\frac{\partial(\mathbf{F}(\mathbf{X}))_u}{\partial\mathbf{x}_v}\right|}. \qquad (2)$$

In particular, when restricted to linear maps, the measures defined in Equations 1 & 2 are the unique measures from Theorems 3.1 & 3.2.

Since a GNN with weights $\Theta$ parameterizes a differentiable map $\mathbf{F}_\Theta(\mathbf{X}) = \mathbf{Y}$, we can straightforwardly use this definition to compute the range of any GNN. Furthermore, we can generalize the measure to the case of multiple input and output channels in an equivalent form as an expectation with respect to the influence distribution $I_u$:

$$\hat{\rho}_u(\mathbf{F}) := \mathbb{E}_{v\sim I_u}\left[d_{\mathsf{G}}(u, v)\right], \qquad (3)$$

where $I_u(v) = \frac{1}{N_u}\sum_{\alpha,\beta}\left|\frac{\partial\mathbf{F}_u^\alpha}{\partial\mathbf{x}_v^\beta}\right|$, normalizing constant $N_u = \sum_{w,\alpha,\beta}\left|\frac{\partial\mathbf{F}_u^\alpha}{\partial\mathbf{x}_w^\beta}\right|$, and $\alpha$ and $\beta$ index the output and input channels respectively. This formulation extends naturally to higher dimensions, since the influence distribution is not restricted to dimension 1. Using this viewpoint, we can interpret $\hat{\rho}_u$ as measuring *the average distance over all interactions*, or as the average displacement after following a random walk defined by the influence distribution. Additionally, the probabilistic viewpoint permits fast stochastic approximations of the normalized range. For these reasons, we favor the normalized range in our experiments.

**Range granularities**. We define a hierarchy of range granularities to capture task-specific statistics at different levels. The *node-level node range* $\hat{\rho}_u$ is defined for every node $u$. Averaging over all nodes in a graph $\mathsf{G}$ yields the *node-level graph range* $\hat{\rho}_{\mathsf{G}}$. Similarly, averaging over all graphs in a dataset $\mathcal{G}$ gives the *node-level dataset range* $\hat{\rho}_{\mathcal{G}}$. This hierarchy allows the range measures to be computed on both transductive and inductive tasks. We use the graph and dataset granularity for our figures and experiments in Sections 5 and 6. See Table 1 for details and a summary.

## 4. Range of a graph-level task

For node-level functions, the Jacobian — corresponding to the first-order term in the Taylor expansion — provides a set of interactions between all nodes, making it suitable to capture pairwise node sensitivity. However, for a graph-level function $\mathbf{y}(\mathbf{X}) \in \mathbb{R}^c$, the Jacobian is a vector, only capturing the sensitivity of the output with respect to individual nodes. This makes it unsuitable for assessing pairwise interactions. To capture such interactions, we use the Hessian, which appears in the second-order term of the Taylor expansion. This is in line with the mixing measure from Giovanni et al. (2024), in which the Hessian is used to study information propagation in GNNs. For completeness, we write down the Taylor expansion of a task in the 1-dimensional case, where $\boldsymbol{\Delta} \in \mathbb{R}^n$ is a small perturbation applied to $\mathbf{X}$:

$$
\mathbf{y}(\mathbf{X} + \boldsymbol{\Delta}) = \mathbf{y}(\mathbf{X}) + \underbrace{\frac{\partial \mathbf{y}(\mathbf{X})}{\partial \mathbf{X}} \boldsymbol{\Delta}}_{\text{first order}}
$$
$$
+ \underbrace{\sum_{u \neq v} \frac{\partial^2 \mathbf{y}(\mathbf{X})}{\partial \mathbf{x}_u \partial \mathbf{x}_v} \boldsymbol{\Delta}_u \boldsymbol{\Delta}_v + \sum_u \frac{\partial^2 \mathbf{y}(\mathbf{X})}{\partial \mathbf{x}_u^2} \frac{1}{2} (\boldsymbol{\Delta}_u)^2}_{\text{second order}}
$$
$$
+ \underbrace{\mathcal{O}\left(\|\boldsymbol{\Delta}\|^2\right)}_{\text{higher order}}
$$

We can then define the graph-level range in the one-dimensional case as the analogous quantity:

$$
\eta_u(\mathbf{y}) := \sum_{v \in \mathsf{V}} \left| \frac{\partial^2 \mathbf{y}}{\partial \mathbf{x}_u \partial \mathbf{x}_v} \right| d_{\mathsf{G}}(u, \, v),
$$

and its normalized version:

$$
\hat{\eta}_u(\mathbf{y}) := \frac{\eta_u(\mathbf{y})}{\sum_{v \in \mathsf{V}} \left| \frac{\partial^2 \mathbf{y}}{\partial \mathbf{x}_u \partial \mathbf{x}_v} \right|} \, . \tag{4}
$$

As in the node-level case, we extend this to higher dimensions for both input and output channels and represent the measure as an expectation:

$$
\hat{\eta}_u(\mathbf{y}) := \mathbb{E}_{v \sim J_u}[d_{\mathsf{G}}(u, v)],
$$

where $J_u$ is the distribution $J_u(v) = \frac{1}{N_u} \sum_{\alpha, \beta, \gamma} \left| \frac{\partial^2 \mathbf{y}^\gamma}{\partial \mathbf{x}_u^\alpha \partial \mathbf{x}_v^\beta} \right|$ with normalizing constant $N_u = \sum_{v, \alpha, \beta, \gamma} \left| \frac{\partial^2 \mathbf{y}^\gamma}{\partial \mathbf{x}_u^\alpha \partial \mathbf{x}_v^\beta} \right|$, and $\alpha, \beta, \gamma$ are the dimensions of the inputs and the output. The term $\frac{\partial^2 \mathbf{y}^\gamma}{\partial \mathbf{x}_u^\alpha \mathbf{x}_v^\beta}$ corresponds to the mixing between channel $\alpha$ of $\mathbf{x}_u$ and channel $\beta$ of $\mathbf{x}_v$ to compute channel $\gamma$ of $\mathbf{y}(\mathbf{X})$, so this quantity measures the total mixing between the two nodes. Similarly to the node-level measure, the normalized range generalizes to higher dimensions, enabling fast stochastic approximations of $\hat{\eta}_u$.

**Range granularities**. As in the node-level case, we define a hierarchy of range granularities for graph-level tasks. The *graph-level node range* $\hat{\eta}_u$ is defined per node $u$. Averaging over nodes in a graph $\mathsf{G}$ yields the *graph-level graph range* $\hat{\eta}_{\mathsf{G}}$, and averaging over graphs in a dataset $\mathcal{G}$ gives the *graph-level dataset range* $\hat{\eta}_{\mathcal{G}}$. See Table 1 for details.

## 5. Task Range Examples

In this section we illustrate the proposed range measures, computing them for different tasks and topologies for node- and graph-level tasks, both empirically and analytically.

### 5.1. Design of synthetic tasks

We design synthetic examples under a general framework of pairwise tasks on graphs. The framework consists of the combination of a *distance* function $\mathcal{D} : \mathsf{V} \times \mathsf{V} \to \mathbb{R}$ and an *interaction function* $\mathcal{I} : \mathbb{R}^d \times \mathbb{R}^d \to \mathbb{R}^c$. A node-level task is obtained as $\mathbf{F}(\mathbf{X})_u = \bigoplus_{v \in \mathsf{V}} \mathcal{D}(u, v)^{-1} \cdot \mathcal{I}(\mathbf{x}_u, \mathbf{x}_v)$, where $\bigoplus$ denotes an aggregation. The interaction function is a general pairwise function and the distance function scales the magnitude of the pairwise interactions. Graph level tasks can then be obtained as $\mathbf{y}(\mathbf{X}) = \bigotimes_{u \in \mathsf{V}} \bigoplus_{v \in \mathsf{V}} \mathcal{D}(u, v)^{-1} \cdot \mathcal{I}(\mathbf{x}_u, \mathbf{x}_v)$, where $\bigotimes$ is another aggregation. This provides a general framework to design tasks on graphs, which we use to design tasks of varying range.

### 5.2. Empirical examples

In order to investigate the impact of graph topology and task definition on the range measure, we consider several distributions of graph topologies and three linear tasks.

**Impact of topology**. To study the impact of the topology of a graph on the range, we consider node-level regression tasks that predict $\tilde{\mathbf{A}}^k \mathbf{X}$ for some $k$ where $\tilde{\mathbf{A}}^k$ is $k$-th power of the symmetric normalized adjacency with self loops, and graph-level regression tasks which consists of the squared difference interaction and the same distance.

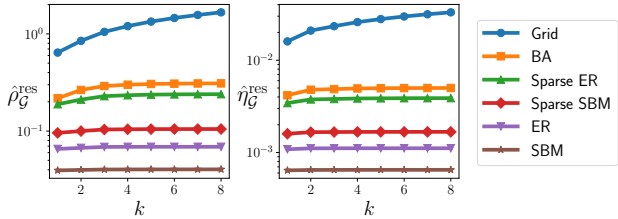

*Figure 2.* **Topology of graph impacts range.** Ranges of node-level tasks predicting $\tilde{\mathbf{A}}^k \mathbf{X}$ (left) and graph-level tasks using a squared difference interaction function and the same distance function, where $\mathcal{D}(u, v)^{-1} = (\tilde{\mathbf{A}}^k)_{uv}$, averaged over nodes (right) for varying $k$ on graph distributions. While task choice affects range, high range requires topologies with sufficiently large diameter.

Figure 2 shows the task range for varying values of $k$ on different distributions of graphs, namely on 1-dimensional grid graphs (i.e. line graphs), on dense ($p = 0.3$) and sparse ($p = 0.1$) Erdős-Renyi graphs (ER) with 100 nodes, and on dense ($p_{\text{intra}} = 0.75$, $p_{\text{inter}} = 0.25$) and sparse ($p_{\text{intra}} = 0.3$, $p_{\text{inter}} = 0.1$) stochastic block models (SBM) with 100 nodes. We notice that, overall, the ranges of the tasks increase with $k$, but the topology affects the *rate* at which the range grows. Indeed, sparse topologies (e.g. grids) that permit large diameters result in larger ranges than those induced by their denser counterparts (ER, SBM).

**Impact of task**. We now fix the topology to be a grid-graph and compute the range of the three following tasks:

$k$-**Power:** Predicting the function $\mathbf{F}(\mathbf{X}) = \tilde{\mathbf{A}}^k \mathbf{X}$ where $\tilde{\mathbf{A}}^k$ is the $k$-th power of the symmetric normalized adjacency matrix without self-loops.

$k$-**Rectangle:** Predicting the function $\mathbf{F}(\mathbf{X})_u = \frac{1}{|\mathcal{N}_{\leq k}(u)|} \sum_{v \in \mathcal{N}_{\leq k}(u)} \mathbf{X}_v$, the mean of inputs over all nodes within $k$ hops.

$k$-**Dirac:** Predicting the function $\mathbf{F}(\mathbf{X})_u = \frac{1}{|\mathcal{N}_k(u)|} \sum_{v \in \mathcal{N}_k(u)} \mathbf{X}_v$, the mean of inputs over all nodes that are *exactly* $k$ hops away.

We plot the ranges for different $k$ for all three tasks in Figure 3. As expected, all tasks become more long-range with increasing $k$, with a clear order in the ranges for each fixed $k$: $k$-Dirac $>$ $k$-Rectangle $>$ $k$-Power. This can be understood by visualizing the shape of the influence distribution of the three tasks (see Figure 3): the mass concentrated furthest away for the Dirac task, spread uniformly for the rectangle task, and concentrated locally for the power task.

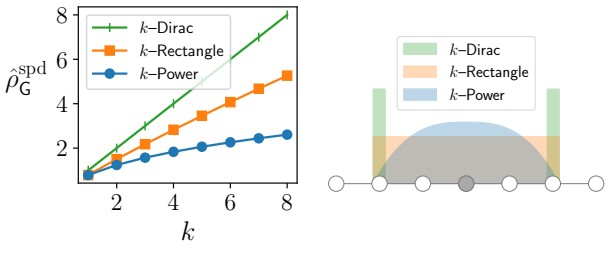

(a) Per-task ranges          (b) Influence distributions

*Figure 3.* **Task impacts range.** (a) illustrates, for each of the 3 synthetic tasks, SPD range on a grid graph for varying $k$. Owing to the shape of each influence distribution, we observe a linear increase for $k$-Dirac and $k$-Rectangle, and a sublinear increase for $k$-Power. (b) illustrates, on a line graph, the influence distribution for the central node for each task when $k = 2$. For $k$-Power the influence decays with distance, while for $k$-Dirac influence is zero everywhere except at the $k$-hop.

### 5.3. Analytic node-level examples

We now consider example tasks whose range can be computed analytically.

**Example 1**. Consider a task that is computable by a node-level function. That is, there exists a $f : \mathbb{R}^d \to \mathbb{R}^c$ such that $\mathbf{F}$ is simply applying $f$ node-wise $\mathbf{F}(\mathbf{X})_u := f(\mathbf{X}_u)$. In this case, no matter the distance metric chosen, $\hat{\rho}_u(F) = \rho_u(F) = 0$, independently of $u$ and $\mathbf{X}$. This shows that in the extreme case of a purely local task, the range is $0$.

**Example 2**. Consider the node-level task which averages the squared difference of node features within the $k$-hop neighborhood, where the node features are sampled i.i.d. from a standard Gaussian distribution:

$$\mathbf{F}^k(\mathbf{X})_u := \frac{1}{|\mathcal{N}_{\leq k}(u)|} \sum_{v \in \mathcal{N}_{\leq k}(u)} (\mathbf{x}_u - \mathbf{x}_v)^2.$$

This is a 1-dimensional task whose Jacobian is given by:

$$\frac{\partial \mathbf{F}^k(\mathbf{X})_u}{\partial \mathbf{x}_v} = \begin{cases} \frac{2}{|\mathcal{N}_{\leq k}(u)|} (\mathbf{x}_v - \mathbf{x}_u) & \text{if } v \in \mathcal{N}_{\leq k}(u) \\ 0 & \text{otherwise,} \end{cases}$$

and whose normalized range with respect to SPD is:

$$\hat{\rho}_u^{\text{spd}}(\mathbf{F}^k) = \frac{1}{N_u} \sum_{r=1}^{k} \sum_{v \in \mathcal{N}_r(u)} \frac{2}{|\mathcal{N}_{\leq k}(u)|} |\mathbf{x}_v - \mathbf{x}_u| \, r,$$

where $N_u := \sum_{v \in \mathcal{N}_{\leq k}(u)} \frac{2}{|\mathcal{N}_{\leq k}(u)|} |\mathbf{x}_v - \mathbf{x}_u|$ is the normalizing constant. Since the features are i.i.d. Gaussian, $|\mathbf{x}_u - \mathbf{x}_v|$ is half-Gaussian with expectation $\frac{2}{\sqrt{\pi}}$, so the expected $N_u$ is $\frac{4}{\sqrt{\pi}}$ and the expected normalized range is:

$$\mathbb{E}_{\mathbf{X}}[\hat{\rho}_u^{\text{spd}}(\mathbf{F}^k)] = \frac{1}{|\mathcal{N}_{\leq k}(u)|} \sum_{r=1}^{k} |\mathcal{N}_r(u)| \, r.$$

In particular, we have $\mathbb{E}_{\mathbf{X}}[\hat{\rho}_u^{\text{spd}}(\mathbf{F}^{k+1})] \geq \mathbb{E}_{\mathbf{X}}[\hat{\rho}_u^{\text{spd}}(\mathbf{F}^k)]$, with strict inequality if $\mathcal{N}_{k+1}(u)$ is non-empty, showing that the range increases with $k$.

### 5.4. Analytic graph-level examples

**Example 1**. Consider a task that is computable without mixing between nodes, namely, there exists a node-wise $f : \mathbb{R}^d \to \mathbb{R}^d$ function such that the task $\mathbf{y}$ is simply a sum over all nodes: $\mathbf{y}(\mathbf{X}) = \sum_u f(\mathbf{x}_u)$. In this case, no matter the metric chosen, $\eta_{\mathsf{G}}(\mathbf{y}) = \hat{\eta}_{\mathsf{G}}(\mathbf{y}) = 0$.

**Example 2**. Consider the graph-level task which sums the squared difference of features on nodes within the $k$-hop neighborhood followed by averaging over the graph:

$$\mathbf{y}^k(\mathbf{X}) := \frac{1}{|\mathsf{V}|} \sum_{u \in \mathsf{V}} \sum_{v \in \mathcal{N}_{\leq k}(u)} (\mathbf{x}_u - \mathbf{x}_v)^2.$$

The second derivatives are $\frac{\partial^2 \mathbf{y}}{\partial \mathbf{x}_u^2} = \frac{4|\mathcal{N}_{\leq k}(u)|}{|V|}$ and:

$$\frac{\partial^2 \mathbf{y}}{\partial \mathbf{x}_u \partial \mathbf{x}_v} = \begin{cases} -\frac{4}{|V|} & \text{if } v \in \mathcal{N}_{\leq k}(u) \\ 0 & \text{otherwise.} \end{cases}$$

Hence the normalizing constant is $N_u = \frac{8|\mathcal{N}_{\leq k}(u)|}{|V|}$, resulting in the following normalized range:

$$\hat{\eta}_u^{\text{spd}}(\mathbf{y}^k) = \frac{1}{2|\mathcal{N}_{\leq k}(u)|} \sum_{r=1}^{k} |\mathcal{N}_r(u)| \, r,$$

showing that the range increases with $k$ as the task incorporates interaction between more distant nodes.

# 6. Experiments

In many real-world scenarios, the underlying task is unknown, making it essential to characterize task properties empirically. In this section, we present empirical results in this direction on both synthetic and real-world experiments. First, we demonstrate that *the range of a trained model that solves a task approximates the range of the underlying task.* We then leverage this observation to analyze the LRGB benchmark by the range of models trained on its tasks.

While this approximation holds in synthetic settings where the task is solved nearly perfectly, real-world models do not achieve zero error, and thus do not provide a unique or exact estimate of task range. Nevertheless, we find that the *correlation* between model range *and performance* across architectures offers a useful heuristic: tasks where better-performing models tend to exhibit longer range are plausibly long-range in nature, while those where performance saturates at low range are likely dominated by local interactions.

### 6.1. Do GNNs approximate the true range of a task?

This section investigates whether the range of a well-trained GNN can approximate the range of the underlying task. When using a GNN, we implicitly assume that the target function is parameterizable and thus admits a true range. This raises a central question: under what conditions does the range of a GNN align with that of the task? We argue that when the validation error approaches zero — i.e., when the model generalizes well — the range of the GNN converges to the task range. We demonstrate this empirically in a controlled synthetic setting, across both node- and graph-level tasks. These results support using the range of a trained GNN as a practical proxy for the range of a real-world task.

Figure 4 illustrates linear GCNs with residual connections of varying depths predicting $\tilde{\mathbf{A}}^5 \mathbf{X}$ on a 1D grid graph — a 5-hop task with $\hat{\rho}_{\mathcal{G}}^{\text{res}} = 1.33$. Models are trained on 500 samples with i.i.d. standard Gaussian node features. We track the evolution of the range of a model during training

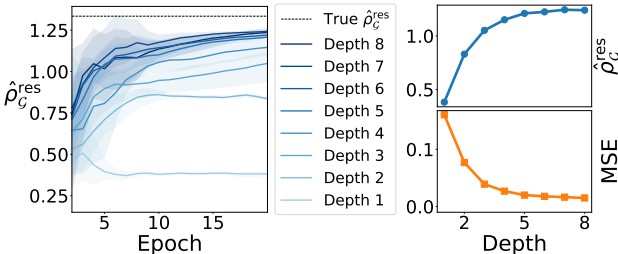

*Figure 4.* **Node-level range can be approximated with a good model.** (Left) Average node-level dataset range evolutions for different GCN depths on a line graph for the 5-power task. Shaded area corresponds to min/max over 4 seeds. (Right) Range and MSE at epoch 20 against number of GCN layers.

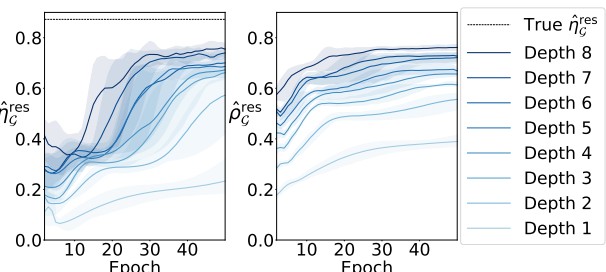

*Figure 5.* Average dataset range evolutions for different GCN depths on a line graph, using the 5-Power distance function and squared difference interaction function from Section 5.4. Shaded area corresponds to min and max over 4 seeds. (Left) **Graph-level range can be approximated with a good model.** (Right) **Pre-pooling node-level range correlates with graph-level range.**

and plot final MSE vs. depth. Deeper models approach the true task range and achieve lower MSE, while shallow models under-perform due to limited receptive fields and under-reaching. These results demonstrate that, under ideal conditions, model range converges toward the true task range as training progresses and depth increases.

We repeat this on a graph-level task as shown in Figure 5. We use the same distance function as for the node-level but replace the interaction function with the squared difference followed by averaging over the nodes, as in Section 5.4. This makes the task non-linear and guarantees a non-trivial Hessian. We also use GeLU activation to make the models non-linear. Similarly to the node-level case, the graph-level range $\hat{\eta}_{\mathcal{G}}^{\text{res}}$ of deeper models tends towards the true task range. Figure 5 also shows the mean node-level range of the pre-pooling node embeddings, $\hat{\rho}_{\mathcal{G}}^{\text{res}}$; we can see that $\hat{\rho}_{\mathcal{G}}^{\text{res}}$ follows a similar trend to $\hat{\eta}_{\mathcal{G}}^{\text{res}}$, suggesting a correlation between the two measures. This justifies using $\hat{\rho}_{\mathcal{G}}^{\text{res}}$ over $\hat{\eta}_{\mathcal{G}}^{\text{res}}$ in our LRGB experiments in Section 6.2, and allows a greater focus more on pairwise interactions induced by message-passing.

## 6.2. Validating real-world benchmarks

The goal of this section is to evaluate (i) whether GNNs trained on existing long-range benchmarks do indeed learn long-range functions, and (ii) whether the underlying benchmark tasks themselves are long-range.

As previously mentioned, we acknowledge that the unsolved nature of real-world tasks means that range of a model alone cannot definitively give us the range of the underlying task; furthermore, models might exhibit bias towards particular ranges depending on their design. To address this, we analyze the *correlation of performance and range* for a spectrum of architectures, to determine if a model is a good approximator, and whether long-range interactions are beneficial to the downstream task.

Furthermore, we include additional experiments on CORA, a known short-range task, and on heterophilic tasks (Platonov et al., 2023) in Appendix C. These experiments support the intuition that the correlation between performance and model range is a suitable heuristic to assess the range of a task, and show that GTs are able to learn to be short-range.

### 6.2.1. EXPERIMENTAL SETUP

**Models**. We consider four models, GCN (Kipf & Welling, 2017), GINE (Xu et al., 2018a; Hu et al., 2020), GatedGCN (Bresson & Laurent, 2017) and GPS (Rampášek et al., 2022), from Tönshoff et al. (2023), which reports more accurate baseline LRGB performance than Dwivedi et al. (2022), a hyperparameter search and correct normalization having been performed for each task. We also use a pure graph transformer without message-passing (GT; Vaswani et al. (2017)) and a GCN with a virtual node (GCN+VN) to ensure a set of models with broad range tendencies.

**Tasks**. From LRGB, we consider VOCSUPERPIXELS, a node-level classification task, and PEPTIDES-FUNC and PEPTIDES-STRUCT, graph-level classification and regression tasks respectively. The models are trained using the hyperparameters from Tönshoff et al. (2023). We track $\hat{\rho}_{\mathcal{G}}^{\text{res}}$ and $\hat{\rho}_{\mathcal{G}}^{\text{spd}}$ throughout training using the Jacobian of node output features with respect to input features. Figures 6 & 7 show evolution of model range over training for a subset of the validation split; 500 graphs for VOCSUPERPIXELS, 200 each for PEPTIDES. We use subsets as we found that this approximates the full dataset ranges well, while being significantly less compute-intensive. We report only validation results as range estimates were found to be highly consistent across splits.

**Jacobian sampling**. To reduce the computational cost of range computation, we use an estimate of the range (see Equation 3) obtained from a sub-sampling of the Jacobian. This is essential due to the large size and feature dimensionality of the LRGB tasks, and we observe that it does

not compromise accuracy. Sub-sampling is performed over output nodes as well as input and output feature channels, according to by pre-defined probability hyperparameters (details in Appendix D.4).

**Distance metric**. We focus on $\hat{\rho}_{\mathcal{G}}^{\text{spd}}$ as (i) SPD is more interpretable than resistance distance, and (ii) results were generally similar to those for $\hat{\rho}_{\mathcal{G}}^{\text{res}}$. In particular, PEPTIDES graphs have similar topologies to line graphs, for which SPD and resistance are equivalent. Figures for resistance range can nevertheless be found in Appendix C.2. VOCSU-PERPIXELS, PEPTIDES-FUNC and PEPTIDES-STRUCT are evaluated on F1 score, average precision (AP) and mean absolute error (MAE) respectively.

**Graph-level tasks**. For the graph-level PEPTIDES tasks, we compute the node-level, Jacobian-based range (see Table 1) using node representations obtained after the final message-passing layer, but before global pooling and final MLP. This is partly due to the high computational cost of estimating the Hessian for large graphs with high feature dimensionality. More importantly, since the post-message-passing components are standard architectural elements applied uniformly across models, we focus on the long-range interactions induced specifically by the *GNN* layers.

### 6.2.2. RESULTS

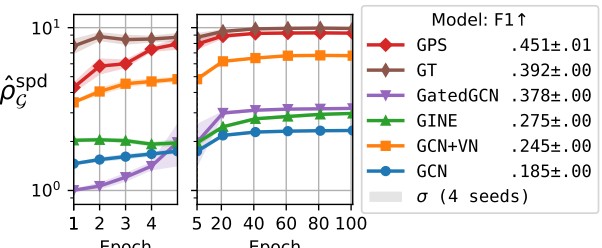

*Figure 6.* $\hat{\rho}_{\mathcal{G}}^{\text{spd}}$ for MPNN and GT models during training, evaluated on a 500-graph subset of the validation split for VOCSUPER-PIXELS. We cut off after 100 epochs as ranges have converged. The fact that $\hat{\rho}_{\mathcal{G}}^{\text{spd}}$ settles at ~2-3 for MPNNs and at ~10 for GPS, and that range positively correlates with performance, suggests that VOCSUPERPIXELS is, to some degree, long-range. Standard deviation is over 4 model/sampling seeds. Log scale on y-axis.

**VOCSUPERPIXELS**. Figure 6 shows that all models increase in range during training, primarily within the first few epochs. MPNNs reach a final range of ~2–3 hops, while GPS and GT reach around ~10. This is consistent with the notion that MPNNs learn local information first and incorporate more distant information as training progresses. Additionally, *range correlates positively with performance* across models, and relative ranges are in line with our understanding of each model.

GINE and GatedGCN, being more expressive than the

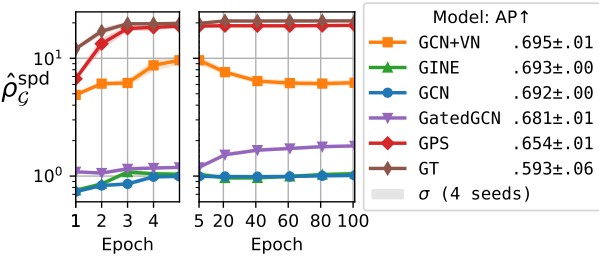

(a) PEPTIDES-FUNC

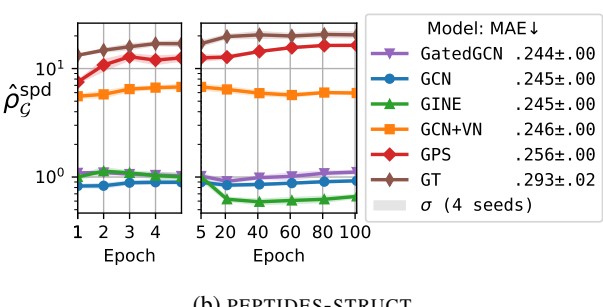

(b) PEPTIDES-STRUCT

*Figure 7.* $\hat{\rho}_{\mathcal{G}}^{\text{spd}}$ for MPNN and GT models during training, evaluated on a 200-graph subset of the validation splits for PEPTIDES-FUNC and PEPTIDES-STRUCT. We cut off after 100 epochs as ranges have converged. The negative correlation between performance and range may suggest that the PEPTIDES tasks are not as long-range as previously assumed. Standard deviation is over 4 model/sampling seeds. Log scale on y-axis.

purely convolutional GCN, improve resistance to the over-smoothing and over-squashing phenomena that hamper long-range interactions, and therefore attain higher ranges — but they are ultimately still MPNNs, so the difference is slight. GCN+VN improves significantly on the performance of a pure GCN and induces a larger range, as the VN mitigates under-reaching by creating a 2-hop connection between all nodes. GPS, the best-performing model, uses global attention — which permits a large range and mitigates over-squashing — alongside a local GatedGCN component.

The observed positive correlation between model range and performance suggests that VOCSUPERPIXELS is inherently a long-range task, in that successfully solving it requires incorporating information from distant nodes. The poor performance of MPNNs on this task can therefore be attributed to their limited ability to capture long-range dependencies, likely due to under-reaching, over-smoothing, and over-squashing.

**PEPTIDES**. In contrast to VOCSUPERPIXELS, PEPTIDES-FUNC and PEPTIDES-STRUCT exhibit the opposite trend: GT and GPS perform the worst while the three MPNNs all perform similarly well. Notably, the MPNNs maintain a range $\hat{\rho}_{\mathcal{G}}^{\text{spd}}$ of ~1 hop after epoch 1, with no increase during

training; for PEPTIDES-STRUCT the range of GINE even *decreases*. This suggests that the PEPTIDES *tasks are inherently local*. Although GPS and GT exhibit larger ranges — and to a lesser extent, GatedGCN on PEPTIDES-FUNC — these do not translate into improved performance. Similarly, GCN+VN has higher range than the MPNNs but yields minimal or no performance gains. These findings indicate that while some models are capable of modeling long-range interactions, such capacity is unnecessary for solving the PEPTIDES tasks effectively.

For PEPTIDES-STRUCT this finding is less surprising; Tönshoff et al. (2023) report that GCN remains state-of-the-art, even when compared against an array of more expressive models. For PEPTIDES-FUNC, however, several later works report improved performance compared to the models considered here. Notably, these approaches often incorporate some form of long-range information flow, such as dynamically rewired multi-hop (Gutteridge et al., 2023) or global attention with added inductive bias (Ma et al., 2023). However, both of these examples emphasize the importance of *short-range interactions* in their architectures by integrating strong local inductive bias. Indeed, this seems to be a requirement for long-range models to perform well on PEPTIDES-FUNC. Given that our findings suggest that PEPTIDES-FUNC may not be especially long-range, these models' performance is, perhaps, more attributable to *how information is propagated* than to their range capabilities.

## 7. Discussion

This work formalizes long-range interactions for graph tasks, introduces a family of principled and quantitative range measures, and applies them to synthetic and real-world tasks. We believe that these measures will serve as tools that afford both a greater understanding of the long-range problem in graph machine learning, and as an additional validation method for future proposed architectures and benchmark tasks. More broadly, our findings highlight the need for GNN evaluation to move beyond performance metrics, incorporating interpretable measures—such as range—for more transparent and principled assessment. Finally, our proposed range measures are not limited to GNNs or graph learning tasks and can be applied to any geometric domain equipped with a notion of distance.

Future work will further investigate our Hessian-based range measures $\hat{\eta}$ for evaluating real-world benchmarks alongside, and in comparison with, the node-level range. Future work will also focus on investigating specific architectural components such as fully connected layers, graph rewiring, residual connections, and positional or structural encoding, and their impact on the range of a model. This will include both empirical and theoretical analysis to assess how common GNN building blocks influence range in a principled way.

## Impact Statement

This paper presents work whose goal is to advance the field of machine learning. There are many potential societal consequences of our work, none which we feel must be specifically highlighted here.

## Acknowledgments

We thank Federico Barbero, İsmail İlkan Ceylan and Francesco Di Giovanni, discussions with whom helped shape the presentation of this work. BG and SLR acknowledge funding support from EPSRC Centre for Doctoral Training in Autonomous Intelligent Machines and Systems No. EP/S024050/1. MB and JB are partially supported by the EPSRC Turing AI World-Leading Research Fellowship No. EP/X040062/1 and EPSRC AI Hub No. EP/Y028872/1. XD acknowledges support from the Oxford-Man Institute of Quantitative Finance and the EPSRC No. EP/T023333/1.

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

# A. Detailed discussion of related work

In this section we review existing attempts at long-range benchmarks, both synthetic and real-world, as well as methods that attempt to solve the long-range problem, and how they validate their claims.

## A.1. Long-range benchmarks

Dwivedi et al. (2022) argue that the standard suite of GNN benchmarks are inappropriate for evaluating long-range interactions as they are overwhelmingly local, consisting of graphs with a small number of nodes and narrow diameter. They then introduce the Long Range Graph Benchmark (LRGB), a suite of tasks whose long-rangedness is established, variably, by (i) large graph size/diameter, (ii) the necessity of graph-level mixing, and (iii) arguments based on an intuitive understanding of the underlying task — e.g. classifying long peptide chains with globally-dependent properties. They also show a performance gap between 'short-range' standard MPNNs and GTs utilizing global attention, though Tönshoff et al. (2023) have since shown that this gap is significantly narrowed by simple hyperparameter tuning, calling into question the extent to which the LRGB tasks are truly long-range. Despite this, LRGB remains the de-facto validator of long-range performance.

Synthetic tasks are also frequently used. Examples include RingTransfer (Bodnar et al., 2021), where a source node must infer a distant target node's label on a ring graph, a color connectivity task where a graph with binary-labelled nodes must be classified as having one 'island' of clustered labels or two (Rampášek & Wolf, 2021), and tasks approximating searches on trees (Lukovnikov et al., 2020; Lukovnikov & Fischer, 2021) such as NeighborsMatch (Alon & Yahav, 2020), and GLoRa (Zhou et al., 2025), which introduces synthetic tasks that require identifying specific long paths in graphs. Several of these tasks bear similarities to tasks from the popular Long Range Arena (Tay et al., 2021) benchmark from the sequence literature; they frequently use very simple topologies (lines, rings, trees), are simple tasks with limited or no node features, and usually depend *only on long-range interactions*, such that a simple rewiring converts them into short-range tasks. Few synthetic tasks acknowledge that graph-structured data and tasks are, by design, locally biased: a task may involve long-range interactions, but will likely be primarily based on local ones. In this paper, we design synthetic experiments that better reflect this fact, requiring long-range interactions without neglecting short ones.

## A.2. Empirical solutions to the long-range problem

Many methods have been proposed for addressing the long-range problem, often with the correlated goal of reducing over-squashing. Simple methods such as adding virtual or latent nodes (Gilmer et al., 2017; Southern et al., 2024; Hariri & Vandergheynst, 2024) or fully adjacent layers (Alon & Yahav, 2020) remove any meaningful distance between nodes by making them all connected within one or two hops, and are often quite effective. Graph rewiring approaches work in a similar fashion, improving the connectedness of nodes with additional edges, typically reducing the graph diameter, either as a pre-processing step (Topping et al., 2021; Gasteiger et al., 2019; Deac et al., 2022; Arnaiz-Rodríguez et al., 2022; Black et al., 2023; Karhadkar et al., 2022; Barbero et al., 2023) or acting on the computational graph within the model architecture (Abu-El-Haija et al., 2019; Abboud et al., 2022; Gutteridge et al., 2023; Bamberger et al., 2025; Finkelshtein et al., 2024). Rewiring has been shown to mitigate over-squashing, at the risk of diluting the benefit of the inductive bias afforded by graph topology. Global attention, such as that used by GTs (Vaswani et al., 2017; Wu et al., 2021; Rampášek et al., 2022), also throws away topology in favor of allowing nodes to interact directly, regardless of distance. Reflecting the inherently local nature of most graph tasks, the best-performing GT architectures tend to be those that combine local with global/long-range components (Ma et al., 2023; He et al., 2023). Increasingly, the relationship between vanishing gradients and the long-range problem is being studied, with inspiration taken from sequence literature: residual connections (Xu et al., 2018b; Gutteridge et al., 2023), gating (Lukovnikov et al., 2020) and state-space models (Choi et al., 2024; Wang et al., 2024; Arroyo et al., 2025), treating the spatial dimension of graphs as analogous to the temporal dimension of sequences.

All of these methods have been argued to improve performance for long-range interactions based on empirical performance on benchmarks. Our work provides an additional and more principled validator by a means of formally measuring the range of a model trained on a given task.

## B. LRGB dataset statistics

In this section we discuss some dataset statistics of interest for VOCSUPERPIXELS and PEPTIDES, which can be found in Table 2. These statistics are taken from Dwivedi et al. (2022), but it should be noted that there exist disconnected graphs in the PEPTIDES datasets, which may affect their accuracy.

*Table 2.* Comparison of VOCSUPERPIXELS and PEPTIDES datasets across various metrics. We report the average validation range over 4 seeds at the final epoch. The graph statistics are reported according to Dwivedi et al. (2022).

| Metric | VOCSUPERPIXELS | PEPTIDES-FUNC |
|---|---|---|
| GCN $\hat{\rho}_{\mathcal{G}}^{\text{spd}}$ | $2.32 \pm 0.098$ | $1.04 \pm 0.030$ |
| GPS $\hat{\rho}_{\mathcal{G}}^{\text{spd}}$ | $9.15 \pm 0.525$ | $19.08 \pm 8.879$ |
| Avg. SPD | $10.74 \pm 0.51$ | $20.89 \pm 9.79$ |
| Avg. diameter | $27.62 \pm 2.13$ | $56.99 \pm 28.72$ |
| Avg. degree | $5.65$ | $2.04$ |

We note that the range of a GPS induces an SPD range approximately close to the average SPD over the graphs in the corresponding datasets. One can conjecture that this is the result of the attention scores being relatively uniform and struggle to capture more complex and local structures within the graph. In the case of VOCSUPERPIXELS, GPS is the best-performing model, hence uniform attention may be desired for the downstream task. PEPTIDES-FUNC is the opposite; the poor performance of GPS and the similarity between SPD and $\hat{\rho}_{\mathcal{G}}^{\text{spd}}$ suggests that the attention mechanism is learning a global averaging rather than complex pairwise interactions. This suggests that the model is simply not well-aligned to the task, and that local structures, which attention is unable to capture, are more important for the downstream task. Further investigation of the influence distributions of these models is required to shed light into the types of structures captured by a model when trained on a specific task.

## C. Additional experimental results

In this Section we include additional experiments and ablations excluded from the main text, on synthetic datasets, on LRGB, CORA and the heterophilic datasets AMAZON-RATINGS and ROMAN-EMPIRE.

### C.1. Additional synthetic experiments

**Node-level range**. In Section 6.1 we look at how well a linear GNN with varying depths can approximate the range of a synthetic task with known true range. We extend this analysis to include a non-linear GNN activation (GeLU) and a GNN with a virtual node. The architecture used for the results in Figure 4 is a GCN (Kipf & Welling, 2017) with no activation plus skip connections. The hidden dimension of the GCN layers is 64. We use $L_2$ loss and a learning rate of 0.001.

**Graph-level range**. We repeat these ablations for the graph-level range in Figure 5 by adding a mean pooling laye in addition to the virtual node. We only include the virtual node ablation here since the original experiment in Figure 5 already has a GeLU activation at every layer to guarantee its ability to capture non-linear interactions. We report the results in Figure 9.

### C.2. Additional LRGB experiments

In this section we include additional figures for resistance distance range experiments for LRGB, which were excluded from the main text due to the similarity of the figures between $\hat{\rho}_{\mathcal{G}}^{\text{spd}}$ and $\hat{\rho}_{\mathcal{G}}^{\text{res}}$. This similarity is especially pronounced for PEPTIDES, as the graphs are very similar to line graphs, for which SPD and resistance distance are equivalent.

### C.3. CORA: a known short-range task

We primarily evaluate range for models trained on LRGB, as it is the de-facto long-range benchmark, but it is desirable to also analyze our range measure on a reference short-range task. We use CORA, and train a graph Transformer and a GCN to compare the range extremities for GNNs. Results are shown in Figure 12. We also show in Figure 13 that, for this task, the GT is capable of learning a similar range to a GCN. This refutes a possible interpretation of our experiments in Section 6.2:

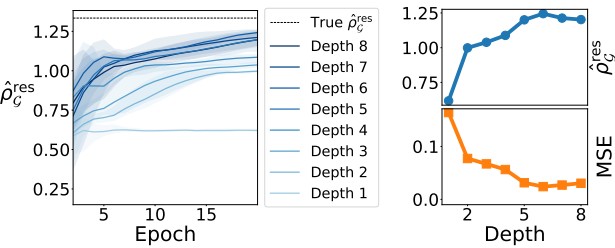 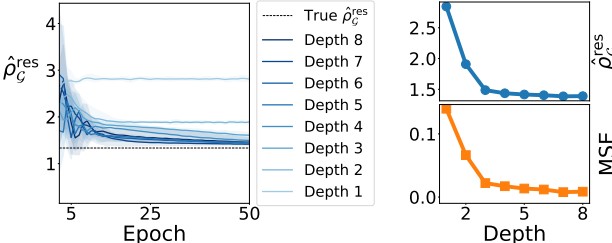

(a) (Left) Average range of **GCN with GeLU activation** for different depths on a line graph for the $k$-Power task for $k = 5$. Shaded area corresponds to min/max over 4 seeds. (Right) Range and MSE at epoch 20 against number of GCN layers.

(b) (Left) Average range of a **GCN+VN** for different depths on a line graph for the $k$-Power task for $k = 5$. Shaded area corresponds to min/max over 4 seeds. (Right) Range and MSE at epoch 50 against number of GCN layers.

*Figure 8.* Node-level range ablations on synthetic $k$-Power task, further validating the range measure on synthetic tasks. Figure 8a shows that including a non-linearity has little effect on range, and Figure 4, whereas adding a virtual node Figure 8b induces more longer ranges at initialization, but this decreases during training as the model successfully learns the task.

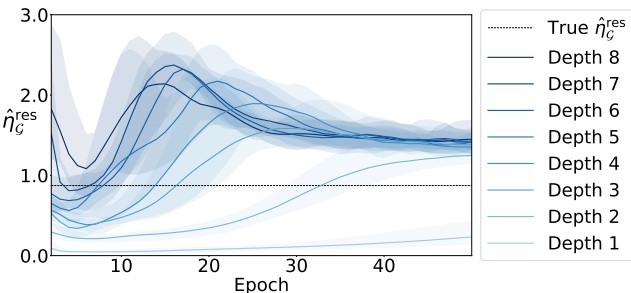

*Figure 9.* Average graph-level range of a **GCN+VN** for different depths on a line graph for the $k$-Power task for $k = 5$. Shaded area corresponds to min/max over 4 seeds. Adding a virtual node produces a similar result to the node-level case; the range is initially higher but trends lower, towards the true range.

that GTs are *always* long-range and are therefore unsuitable for providing information about the underlying range of the task. For these experiments both models use a hidden dimension of 64 and 2 layers, following the hyperparameterizations from Kipf & Welling (2017).

### C.4. AMAZON-RATINGS and ROMAN-EMPIRE: heterophilic tasks

We now investigate if there is a relationship between the long-range interaction problem and heterophilic tasks (Arnaiz-Rodríguez & Errica, 2025). We use our range measure to evaluate models trained on the heterophilic datasets AMAZON-RATINGS and ROMAN-EMPIRE (Platonov et al., 2023). Due to the size of these graphs, 24492 and 22662 nodes for AMAZON-RATINGS and ROMAN-EMPIRE respectively, dense GTs are computationally infeasible. Hence, we test using a local MPNN (GCN), local attention (GT; the same architecture referred to as a GT by Platonov et al. (2023)), and an MPNN with a virtual node (GCN+VN).

Figures 14 & 15 show that only the MPNN+VN is long-range (due to the drastic increase in receptive field) but yields zero performance benefit over a standard GCN; far more important is *how information is propagated*. The local GT, i.e. an MPNN using Transformer-style attention, performs better for both tasks, while range is near-identical to that of a GCN. This suggests that heterophily is not necessarily related to range, at least for these tasks.

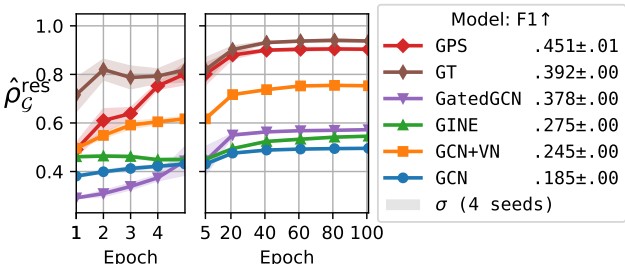

*Figure 10.* $\hat{\rho}_{\mathcal{G}}^{\text{res}}$ for MPNN and GT models during training, evaluated on a 500-graph subset of the validation split for VOCSUPERPIXELS. We cut off after 100 epochs as ranges have converged. The relative range evolutions between models and correlation with performance are very similar to the SPD case, reinforcing our earlier argument about the validity of VOCSUPERPIXELS as a long-range benchmark. However, we note that all ranges are contracted to below 1, even for the GPS and GT. This is due to the topology of VOCSUPERPIXELS graphs (see Table 2), which have a high average degree (see Table 2), meaning that resistance distance grows sub-linearly against SPD. Additionally, the range of GT matches that of GINE and GCN+VN at epoch 1, which contrasts with the SPD range where GT is considerably higher. This confirms the importance of the choice of distance metric. Standard deviation is over 4 model/sampling seeds.

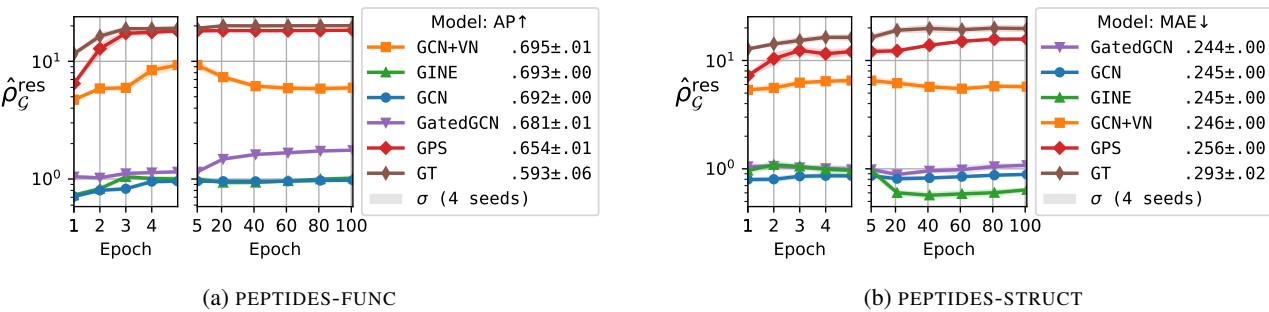

(a) PEPTIDES-FUNC

(b) PEPTIDES-STRUCT

*Figure 11.* $\hat{\rho}_{\mathcal{G}}^{\text{res}}$ for MPNN and GT models during training, evaluated on a 200-graph subset of the validation splits for PEPTIDES-FUNC and PEPTIDES-STRUCT. We cut off after 100 epochs as ranges have converged. The negative correlation between performance and range may suggest that the PEPTIDES tasks are not as long-range as previously assumed. Standard deviation is over 4 model/sampling seeds. We use log scaling on the y-axis.

## D. Additional experimental details

In this section we discuss some additional experimental details excluded from the main text.

### D.1. Differentiability of models for PEPTIDES experiments

Node features for PEPTIDES are integer-valued as they represent atomic features. The `AtomEncoder` module (from GraphGym) used to encode these discrete values uses a `torch.nn.Embedding` module directly. This process prohibits differentiating the model output with respect to the input features, which is necessary for the Jacobian calculation required to compute the range. Hence, we replace the `AtomEncoder` with a `DifferentiableAtomEncoder` which pre-processes the PEPTIDES node features into one-hot floating point vectors. This produces a differentiable input and encoder equivalent to the existing `AtomEncoder` and allows us to compute the Jacobian for our range measures.

### D.2. Inefficiency of one-hot encoding for PEPTIDES

The standard `AtomEncoder` from GraphGym used for PEPTIDES that one-hot encodes integer node features is extremely inefficient: it converts 9 integer-valued node features into a 174-dimensional one-hot vector, but *over 80% of these are zero for all graphs in the* PEPTIDES *dataset*. Only 31 one-hot features are required in practice.

As our range measure involves computing the Jacobian between input and output features, and the complexity of this computation scales with the input feature dimension, we reduce the one-hot encoding to 31 for our `DifferentiableAtomEncoder`. This means our model architectures are slightly different than those reported by

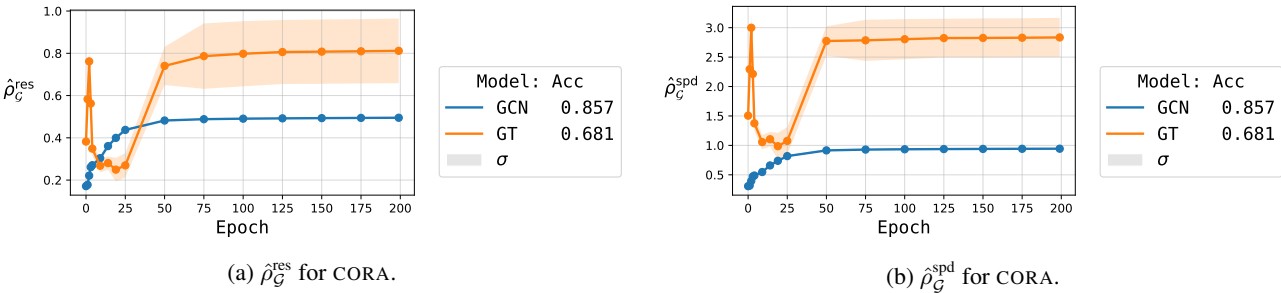

(a) $\hat{\rho}_{\mathcal{G}}^{\text{res}}$ for CORA.

(b) $\hat{\rho}_{\mathcal{G}}^{\text{spd}}$ for CORA.

*Figure 12.* Range experiments on CORA for a GCN and GT. $\hat{\rho}_{\mathcal{G}}^{\text{spd}}$ of the GCN stays at ~1 hop and outperforms the GT, as we would expect from a known short-range task. We can see that in the initial epochs, the GT is able to learn to be more short-range after a high-range initialization; see Figure 13.

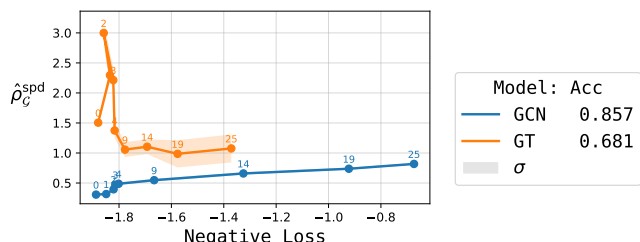

*Figure 13.* Range against loss during training for the test split of CORA. Each point is labelled with its epoch. We cut off after 25 epochs because both models achieved their highest validation accuracy at epoch 25; after this, the GT overfits, as the spiking range from epoch 25 to 50 in Figure 12 shows. We can see that, though the GT learns a high range in the very early epochs, it then *learns a low range*, around 1 hop, before it overfits and its accuracy drops.

Dwivedi et al. (2022); Tönshoff et al. (2023), as the encoding requires fewer parameters, but we found this to have minimal impact on model performance.

We report the updated parameter counts in Table 4.

### D.3. Isolated nodes in PEPTIDES

We discovered that the PEPTIDES dataset contains graphs which are disconnected and may have isolated nodes. To the best of our knowledge, this is not reported anywhere in the literature. Though these graphs make up only ~1% of the dataset, we still consider it an issue. Firstly, the existence of disconnected nodes means the reported results of average graph diameter are technically erroneous; this value would be infinite. Secondly, many methods propose computing metrics on graphs which assume connectivity such as distance metrics of SPD and effective resistance. Moreover, graph Transformer architectures will induce an edge between components of the graph which could never interact in an MPNN, regardless of depth.

To address this, when computing our *range measures* on these graphs with graph Transformer architectures, we initialize distance values to zero and fill the distances by computing them on the respective connected components.

### D.4. Jacobian sampling for range measure estimation

As mentioned in the main text, we sub-sample the Jacobian matrix in order to provide computational speed-ups and avoid out-of-memory issues when computing range measures for real-world experiments. We achieve this by randomly sampling input/output nodes and input/output channels with given probabilities. The unselected node/channels are masked, producing a sparsified Jacobian which results in computational advantage in both time and memory while producing an accurate approximation of the range. This is evident in our experiments, where we report multiple seeds across which the resulting range estimates have extremely low variance. We list sampling parameters used in our experiments in Table 3.

**Sampling hyperparameters**. In Section 3 we present the normalized range as an expectation over the influence distribution. When computing the node-level range for a graph, $\hat{\rho}_{\text{G}}$, this becomes an expectation over input/output nodes and input/output

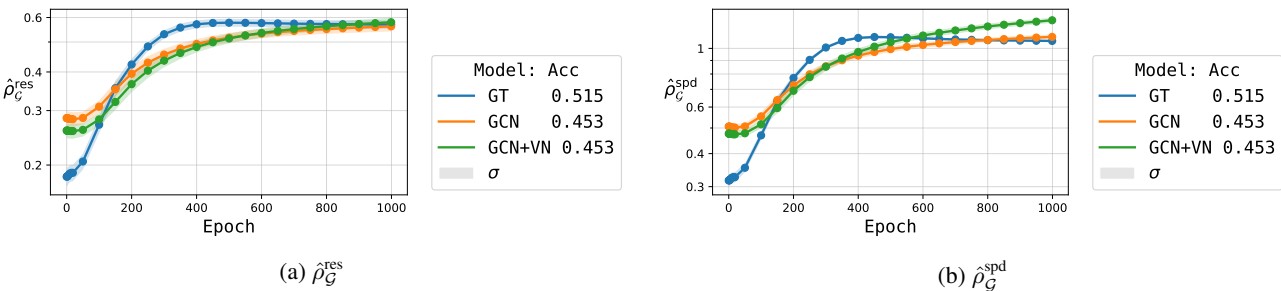

*Figure 14.* Range measures for the AMAZON-RATINGS dataset. There is a clear performance gain from using an attention-based MPNN over convolution. GCN+VN induces slightly more long-range interactions but offers no performance gain, suggesting that AMAZON-RATINGS may *not* be long-range.

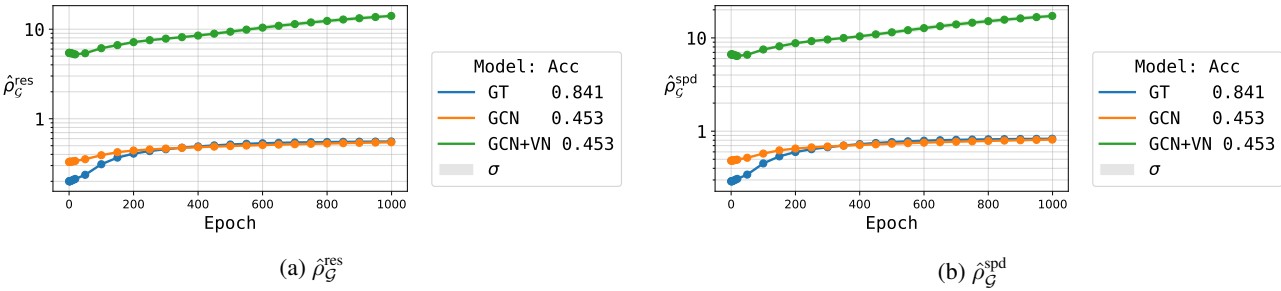

*Figure 15.* Range measures for the ROMAN-EMPIRE dataset. There is a clear performance gain from using an attention-based MPNN over convolution. GCN+VN induces long-range interactions but offers no performance gain, suggesting that ROMAN-EMPIRE may *not* be long-range.

channels. Using a sub-sampling of these allows us to approximate this expectation.

If we assign the probabilities to be $p^{\text{node-in}}, p^{\text{node-out}}, p^{\text{channel-in}}, p^{\text{channel-out}}$, then we reduce the Jacobian size by a factor of $p^{\text{node-in}} \times p^{\text{node-out}} \times p^{\text{channel-in}} \times p^{\text{channel-out}}$.

We construct a binary mask for each component $\mathbf{m} \in \{0, 1\}^n$ such that each entry is independently sampled from a Bernoulli distribution with probability $p$, i.e., $\mathbb{E}\left[\frac{1}{n}\sum_{i=1}^{n} m_i\right] = p$.

We apply the GNN model to the masked input to compute the full output, which is subsequently reduced using the output mask. This selective computation results in a reduced Jacobian and computational graph, offering protection against out-of-memory errors and yielding significant efficiency gains.

Additionally, we found that graphs with $n > \sim 256$ can exceed the memory constraints of an A10 NVIDIA GPU (24GB). Hence, we set an upper bound of max nodes $= 256$. We also set a lower bound of min nodes $= 16$, to ensure we don't under-sample smaller graphs.

### D.5. Computational Resources

Preprocessing, training and range calculations were done on in-house CPUs and GPUs. All experiments were feasible and primarily performed on NVIDIA A10s. Some experiments were performed on NVIDIA H100s.

*Table 3.* Sampling probability hyperparameters used for range experiments. We use a probability of 1.0 for input nodes for all experiments as we found that it introduced bias to the range and did not reduce compute requirements.

| Dataset | Input Nodes | Output Nodes | Input Channels | Output Channels |
|---|---|---|---|---|
| VOCSUPERPIXELS | 1.0 | 0.5 | 0.5 | 0.5 |
| PEPTIDES-FUNC | 1.0 | 0.5 | 0.5 | 0.5 |
| PEPTIDES-STRUCT | 1.0 | 0.5 | 0.5 | 0.5 |
| AMAZON-RATINGS | 1.0 | 0.01 | 1.0 | 0.8 |
| ROMAN-EMPIRE | 1.0 | 0.01 | 1.0 | 1.0 |
| CORA | 1.0 | 0.2 | 0.2 | 1.0 |

*Table 4.* Hyperparameters for LRGB experiments in Section 6.2.2. The #Param. row in a) and (b) lists the original parameter count from Tönshoff et al. (2023) in parentheses alongside our decreased parameter account due to efficient one-hot encodings (see Appendix D).

(a) Hyperparameters for PEPTIDES-FUNC experiments

|  | GCN | GINE | GatedGCN | GPS | GCN+VN | GT |
|---|---|---|---|---|---|---|
| lr | 0.001 | 0.001 | 0.001 | 0.001 | 0.001 | 0.001 |
| dropout | 0.1 | 0.1 | 0.1 | 0.1 | 0.1 | 0.1 |
| #layers | 6 | 8 | 10 | 6 | 6 | 6 |
| hidden dim. | 235 | 160 | 95 | 76 | 235 | 76 |
| head depth | 3 | 3 | 3 | 2 | 3 | 2 |
| PE/SE | RWSE | RWSE | RWSE | LapPE | RSWE | LapPE |
| batch size | 200 | 200 | 200 | 200 | 200 | 200 |
| #epochs | 250 | 250 | 250 | 250 | 250 | 250 |
| norm | - | - | - | BatchNorm | - | BatchNorm |
| MPNN | - | - | - | GatedGCN | GCN | - |
| #Param. | 456k (486k) | 472k (491k) | 483k (493k) | 470k (479k) | 456k | 464k |

(b) Hyperparameters for PEPTIDES-STRUCT experiments

|  | GCN | GINE | GatedGCN | GPS | GCN+VN | GT |
|---|---|---|---|---|---|---|
| lr | 0.001 | 0.001 | 0.001 | 0.001 | 0.001 | 0.001 |
| dropout | 0.1 | 0.1 | 0.1 | 0.1 | 0.1 | 0.1 |
| #layers | 6 | 10 | 8 | 8 | 6 | 8 |
| hidden dim. | 235 | 145 | 100 | 64 | 235 | 64 |
| head depth | 3 | 3 | 3 | 2 | 3 | 2 |
| PE/SE | LapPE | LapPE | LapPE | LapPE | LapPE | LapPE |
| batch size | 200 | 200 | 200 | 200 | 200 | 200 |
| #epochs | 250 | 250 | 250 | 250 | 250 | 250 |
| norm | - | - | - | BatchNorm | - | BatchNorm |
| MPNN | - | - | - | GatedGCN | GCN | - |
| #Param. | 457k (488k) | 473k (492k) | 433k (445k) | 445k (452k) | 457k | 472k |

(c) Hyperparameters for VOCSUPERPIXELS experiments

|  | GCN | GINE | GatedGCN | GPS | GCN+VN | GT |
|---|---|---|---|---|---|---|
| lr | 0.001 | 0.001 | 0.001 | 0.001 | 0.001 | 0.001 |
| dropout | 0.0 | 0.2 | 0.2 | 0.3 | 0.0 | 0.1 |
| #layers | 10 | 10 | 10 | 8 | 10 | 8 |
| hidden dim. | 200 | 145 | 95 | 68 | 200 | 84 |
| head depth | 3 | 3 | 3 | 2 | 3 | 2 |
| PE/SE | RWSE | none | none | LapPE | RWSE | LapPE |
| batch size | 50 | 50 | 50 | 50 | 50 | 50 |
| #epochs | 200 | 200 | 200 | 200 | 200 | 200 |
| norm | - | - | - | BatchNorm | - | BatchNorm |
| MPNN | - | - | - | GatedGCN | GCN | - |
| #Param. | 490k | 450k | 473k | 501k | 490k | 471k |

