# OpenReview forum: "On Measuring Long-Range Interactions in Graph Neural Networks"
_ICML.cc/2025/Conference — ICML 2025 poster_

### Official Review · Reviewer_ztSX · 2025-02-24

**Overall Recommendation:** 3

**Summary:**

The Long Range Graph Benchmark (LRGB) is a widely adopted tool for evaluating the long-range capabilities of frameworks in long-range graph tasks. This paper identifies its limitations and introduces a formal range measure for operators on graphs, encompassing both node-level and graph-level tasks. Furthermore, in light of the current simplicity of synthetic tasks (which predominantly rely on long-range interactions) ,this paper propose a redesign of graph synthesis tasks. Experimental results demonstrate that the method proposed in this paper provides a better understanding of long-range task in graph machine learning.

**Claims And Evidence:**

The claims in the material are supported by clear and compelling evidence.

**Essential References Not Discussed:**

This paper primarily introduces enhancements to the previous graph long-range evaluation tool, LRGB, addressing its limitations. It constitutes a progressive advancement in the field.

**Experimental Designs Or Analyses:**

The experimental design is conducted on both synthetic and real-world datasets, and the validity of the proposed framework is demonstrated through theoretical analysis and foundational experiments, making the findings convincing.

**Methods And Evaluation Criteria:**

The graph long-range evaluation metric proposed in this paper has been validated on both synthetic and real-world datasets, demonstrating its practical significance.

**Other Comments Or Suggestions:**

## Update for Rebuttal
I confirm that I have read the author response, I will keep my score

**Other Strengths And Weaknesses:**

Strengths:
(1)	The paper is written clearly, and the problem is described effectively.
(2)	The theoretical foundation is well-established, and the theoretical and experimental validations are comprehensive and convincing.

Weaknesses:
(1)	This paper primarily proposes a new method for evaluating long-range dependencies in graphs. However, there are concerns regarding the true extent and scope of its impact.

**Questions For Authors:**

refer to weakness

**Relation To Broader Scientific Literature:**

This paper primarily proposes improvements to the previous graph long-range evaluation tool, LRGB, addressing its limitations. It represents a progressive advancement in the field.

**Theoretical Claims:**

This paper provides clear definitions and analyses of evaluation metrics at the graph level and node level, the scope of their influence, and relevant examples. The corresponding theoretical claims are largely accurate.

---

> ### Author Rebuttal · Authors · 2025-04-01
>
> We thank the reviewer for their positive feedback and recognition of the practical significance and credibility of our work. We address the reviewer’s concerns below. We also note that, following suggestions from other reviewers, we have performed additional experiments for GCN and GT on Cora (as a known short-range dataset to compare with LRGB), virtual node and activation function ablations for our synthetic task corresponding to Fig. 4, a VN and pure Transformer ablation for LRGB, and experiments on the heterophilous tasks Roman-Empire and Amazon-Ratings for a GCN and GT. All additional experiments will be included in the final paper. We discuss these results in the relevant reviewer responses, and would be grateful if the reviewer would consider reading these responses and examining the figures, which are compiled in a pdf at the following link: https://fuchsia-lina-52.tiiny.site/
>
>
> >*“This paper primarily introduces enhancements to the previous graph long-range evaluation tool, LRGB, addressing its limitations…there are concerns regarding the true extent and scope of its impact”*
>
> We thank the reviewer for their feedback. While assessment of LRGB is indeed one of the aims of our work, we would like to politely emphasise that our contributions are far more general and go significantly beyond testing the appropriateness of an existing benchmark.
> Firstly, our measure can be applied to any model trained on any task; it not only enhances the usefulness of LRGB in assessing range but *any graph learning benchmark* — for example, see our additional experiments, in which we assess the range of Cora as well as two heterophilous tasks.
>
> Secondly, LRGB is purely an empirical evaluation method based on performance gap. Our range measure provides a theoretically robust range measurement tool for *any model trained on any task* that can be considered in addition to (or in the absence of) performance gap. We believe this is a significant theoretical contribution that paves the way for further analysis in this direction.
>
> Thirdly, while we use and motivate our range measure only for graph ML tasks, as defined it is a general measure applicable to any operator (including any neural network), and for any data structure for which one can define a distance metric. As a result our method can be applied beyond graphs to the general ML setting, as has been done with other works such as [1]. One example which we leave for future work is a theory-driven evaluation of long-context issues observed for Transformers operating on sequence data, such as is observed with the ‘lost in the middle’ problem [2].
>
> ---
>
> We appreciate the reviewer’s thoughtful feedback and hope that our clarifications have effectively conveyed the broader impact and significance of our work. We are, of course, happy to address any further concerns if the reviewer wishes to elaborate on them. Given the reviewer’s positive comments about our work, our above clarifications, and the significant additional experiments we have provided to strengthen the final version, we would be grateful if the reviewer would consider raising their score.
>
> [1] Barbero, Federico, et al. "Transformers need glasses! information over-squashing in language tasks." Advances in Neural Information Processing Systems 37 (2024): 98111-98142. https://arxiv.org/abs/2406.04267
>
> [2] Liu, Nelson F., et al. "Lost in the middle: How language models use long contexts." Transactions of the Association for Computational Linguistics 12 (2024): 157-173. https://arxiv.org/abs/2307.03172

---

### Official Review · Reviewer_7Dgb · 2025-03-12

**Overall Recommendation:** 5

**Summary:**

The paper works in the area of long-range dependencies for graph neural networks. Its main contribution is to define a new metric to measure the range of a task. This metric can also be applied to GNNs that are trained on a task, approximating the true range of the task. Experiments on the LRGB datasets indicate that PASCAL-VOC is indeed a long-range task while the peptides tasks are not.

**Claims And Evidence:**

Almost all claims are thoroughly supported by evidence. Only the claim that high values in GPS indicate higher long-rangedness of the problem is not properly validated as it could very well be that GPS is always long-ranged (even if it does not need to) while MPNNs are only long-ranged if needed. While I generally agree with the claim, I believe that GPS behaves oddly in the experiments and should thus be checked further.

**Essential References Not Discussed:**

I would have expected to find Gilmer et al as the citation for virtual nodes and not two 2024 papers. The paper is cited, but in an unexpected context (for chemistry, not for virtual nodes).

**Experimental Designs Or Analyses:**

I did not run the code myself. The experimental design mostly makes sense, except that I would like a few additional experiments to strengthen the claims about the LRGB datasets. It would also be nice to see whether Figure 4 would look the same for networks that do use activation functions and maybe even virtual nodes.

**Methods And Evaluation Criteria:**

Yes, the proposed methods and evaluations make a lot of sense. It would still be nice to compare to some (known) short-range datasets to see the difference more pronounced.

**Other Comments Or Suggestions:**

p1c2: I would have expected virtual nodes to be discussed here as well as it is an effective way to enable long-range interactions.
p2c2: I would have liked a bit more intuition of when the range should be large and when it should be small. And I found the example given there not super helpful as it is quite technical without having an idea what we actually want to achieve, but that might be personal.
p3c2: It would be nice to state more explicitly that the "Range of a GNN" is the effective range of what the trained GNN computes (and not something that only depends on the architecture)
p3c2, above eq1: I believe that the connection between Jacobian and best linear approximation is not universally known. (At least I had to stop there for a moment while reading). Since you later mention Taylor anyway, it might be a good idea to mention it here as well since its the first-order Taylor term.
p4c1: It would be nice to mention in which context the influence distribution has been defined (as it is not another way of defining range)
p5c1 k-Power: is this with or without self-loops (I would guess without, but it would be nice to be explicit)

Figure 6/7: Since the range is not that large, it might be nice to write the values as 0.3, 0.6, 1, 10 etc instead of the scientific notation which is a lot slower to parse.

p4c1 L187: reason -> reasons
p4c2 L209: Erdős (with that weird o) and L213 increase->increases
p5c1 L250: Figure 2 -> Figure 3. Also Figure 3 has left/right which should be mentioned in the paragraph.
Figure 3 caption: _a_ linear increase ... and _a_ sublinear increase

**Other Strengths And Weaknesses:**

The paper is really well-written and easy to read! It is also thoroughly proofread which I greatly appreciate.

**Questions For Authors:**

My main questions are about the experiments and resulting claims which are already in the review above, but I try to concisely put them here too:
Q1) Does Figure 4 work similarly for GCN with activation functions and possibly VN?
Q2) what happens to fig 4/5 for GPS? Will that behave oddly?
Q3) Is there a way to exclude the alternative claim "GPS is always more long-range, even if it is not needed" which could also explain the figures. A possible way to check that would be using just a transformer without message passing and/or to run everything on one or two datasets that are known to be short-range (e.g. Cora). Effectively: while I agree with the claim, I think using GPS here does not help without additional experiments for e.g. virtual nodes and pure transformers, as well as "baseline" results of GPS on known short-range tasks for calibration. Would it be possible to add those experiments to make the claim stronger?

**Relation To Broader Scientific Literature:**

The new range metric is novel and a great addition to the arsenal that we have to evaluate what a GNN actually does as well as to evaluate what a graph task actually requires.

**Theoretical Claims:**

The proofs are extremely straightforward and easy to read, and even included in the main paper. I find this very refreshing and very positive.

---

> ### Author Rebuttal · Authors · 2025-04-01
>
> We thank the reviewer for their detailed feedback, and for appreciating the novelty of our contribution and its significance as an evaluation tool for GNNs and graph tasks. We address their concerns below. As suggested, we have performed additional experiments for GCN and GT on Cora (as a known short-range task to compare with LRGB), as well as virtual node and activation function ablations for our synthetic task corresponding to Fig. 4, and a VN and pure Transformer ablation for LRGB. We also performed additional experiments, as suggested by other reviewers, for the heterophilous tasks Roman-Empire and Amazon-Ratings for a GCN and GT. All additional experiments will be included in the final paper. We discuss these results in the relevant reviewer responses; the figures are compiled in a pdf here: https://fuchsia-lina-52.tiiny.site/
>
> >*“Q1) Does Figure 4 work similarly for GCN with activation functions and possibly VN?”*
>
> We have extended the synthetic experiment to produce the equivalent of Fig. 4 for both a GCN with GeLU activation, and a GCN with a VN. We report the results in the linked pdf, Figs. A8-9. We notice that adding a nonlinearity does not considerably change the results. Adding a VN, on the other hand, does: where Fig. 4 showed models with an initial negative range bias before converging during training towards the true range from *below*, the addition of a VN appears to induce a positive bias, so models converge to the true range from above — though they do still converge, i.e. the trained model range still approximates the true task range.
>
> >*“Q2) What happens to fig 4/5 for GPS?”*
>
> We performed this experiment and found that the GPS architecture is not well-aligned with this synthetic task: it trains irregularly and attains poor performance. As a result, there is no clear interpretation of the range of the model on this task. This points to the importance of inferring conclusions about tasks by an examination of both the range and performance across different models (see further discussion below). An analysis of the relationship between performance and range of different Transformer architectures on different tasks is an interesting direction which we leave for future work.
>
> >*“Q3) Is there a way to exclude the alternative claim 'GPS is always more long-range, even if it is not needed' which could also explain the figures”*
>
> We agree with the reviewer that high range scores for GPS are not sufficient, by themselves, to indicate that a task is long-range. This is clear from Figs. 6-7 in the paper, from which we conclude that the Peptides tasks are not long-range while VOCSuperpixels (in relative terms) is, despite very similar, high range scores for GPS. We base our claim about the long-rangedness of VOC on two factors:
>
> (i) that the *MPNNs*, rather than GPS, have higher relative range (as the reviewer points out, MPNNs are long-range *if needed*, suggesting that for VOC, longer range is necessary);
>
> (ii) that the range gap between GPS and MPNNs is accompanied by a performance gap, indicating that models with higher range are better suited to the task.
>
> Neither of these is the case for Peptides. To summarise, it may be that GPS will have a high range for a task when it is not required, but by comparing *relative range* as well as *relative performance over multiple models*, we can make an assessment about the underlying task.
>
> To address the second part of the comment, while we do find that GPS and other GTs are biased towards higher range, and often are long-range when it is not needed, they can learn to be short range. See Fig. A4, in which we plot range against validation loss for the first 25 epochs of training a GCN and GT on Cora, a known short-range task. We see that the GT grows rapidly long-range in early epochs, but quickly learns to be short-range until it achieves its maximum validation accuracy at epoch 25 (after which it overfits). This excludes the alternative claim.
>
> We follow the reviewer’s suggestions for additional experiments to better validate our claims. In addition to Cora, we perform experiments for a GT (GPS without MPNN component) and GCN with a virtual node for the three LRGB tasks; see Figs. A5-7. The results support our findings: GT performs very similarly to GPS, and +VN increases range, with a corresponding increase in performance for VOC but not for Peptides, consistent with our conclusions about each task’s range.
>
> >*“k-Power: is this with or without self-loops?”*
>
> Without; we will clarify this in the final version.
>
> ---
> We thank the reviewer again for their positive and useful feedback, as well as for their other detailed suggestions, which will be added for the final version. We are happy to address any further concerns. Given the reviewer’s comments about the novelty and utility of our work, and the significant additional experiments we have provided to strengthen the final version, we would be grateful if the reviewer would consider raising their score.

---

> > ### Comment · Reviewer_7Dgb · 2025-04-02
> >
> > Thanks a lot for the extensive and helpful answers and the additional experiments which (from my perspective) further strengthen the already strong paper. I have thus adjusted my score.

---

> > > ### Author Response · Authors · 2025-04-05
> > >
> > > We are grateful for the reviewer's appreciation of our responses and additional experiments, and thank them for raising their score. We also thank them again for their time, insightful feedback, and engagement with our work.

---

### Official Review · Reviewer_T8Ph · 2025-03-15

**Overall Recommendation:** 3

**Summary:**

This paper introduces a formal measure for evaluating long-range interactions in Graph Neural Networks (GNNs), addressing the limitations of existing empirical benchmarks like LRGB, which lack theoretical grounding. The proposed measure quantifies a model’s ability to capture long-range dependencies, validated through synthetic experiments and applied to assess common benchmarks. This work provides a principled framework for studying and improving long-range interactions in GNNs.

**Claims And Evidence:**

- It is not clear why Jacobian is used for node-level task while Hessian is used for graph-level task. Could you please clarify?

**Essential References Not Discussed:**

N.A

**Experimental Designs Or Analyses:**

Heterophilous datasets are known to benefit from long-range interactions. It would be valuable to empirically analyze the range of heterophilous tasks to better understand their dependence on long-range dependencies. I suggest the authors examine datasets such as Roman-Empire and Amazon-Ratings [1] to provide deeper insights into how their proposed measure applies to heterophilous settings.

[1] Platonov, O., Kuznedelev, D., Diskin, M., Babenko, A., & Prokhorenkova, L. (2023). A critical look at the evaluation of GNNs under heterophily: Are we really making progress?. arXiv preprint arXiv:2302.11640.

**Methods And Evaluation Criteria:**

The authors claim that the range of a trained model that solves a task approximates the range of the underlying task. However, "solving a task" is not clearly defined. In Section 6.2, where experiments are conducted on real-world datasets, the authors evaluate four models, but none achieve perfect performance. This raises an important question: Is the model failing to capture long-range interactions, or does the task itself not require them? I suggest the authors refine their claim to be more rigorous and explicitly clarify the criteria for determining when a task is truly "solved."

**Other Comments Or Suggestions:**

## update after rebuttal

I have read the rebuttal and would like to keep my score

**Other Strengths And Weaknesses:**

The adaptation of influence scores to define a range measure is an interesting approach. Additionally, the findings on LRGB, particularly that existing methods may not be effectively learning from long-range interactions is interesting.

**Questions For Authors:**

Why use Jacobian for node-level task and Hessian for graph-level task? Can you explain in more details for better clarity?

**Relation To Broader Scientific Literature:**

The proposed method is based on influence scores and adapts them to measure the influence from nodes at different ranges. I suggest the authors discuss how this concept aligns with related work in the broader literature, such as [1], to provide readers with a more comprehensive understanding of existing approaches and how their method fits within this context.

[1] Koh, P. W., & Liang, P. (2017, July). Understanding black-box predictions via influence functions. In International conference on machine learning (pp. 1885-1894). PMLR.

**Theoretical Claims:**

I did not check the proofs comprehensively.

---

> ### Author Rebuttal · Authors · 2025-04-01
>
> We thank the reviewer for their detailed feedback and for appreciating our contributions. We address their concerns below. As suggested, we have performed additional experiments for heterophilous tasks, as well as on Cora, virtual node and activation function ablations for our Fig. 4 synthetic task, and a VN and GT ablation for LRGB. All additional experiments will be included in the final paper. We discuss these results in the relevant reviewer responses; the figures are included in a pdf here: https://fuchsia-lina-52.tiiny.site/
>
> >*“It is not clear why Jacobian is used for node-level task while Hessian is used for graph-level task.”*
>
> Both our node- and graph-level range measures require a term that captures pairwise interactions between nodes. For node-level tasks, the $(i,j)$-th element of the Jacobian represents the sensitivity of node $i$’s output features to node $j$’s input features. For graph-level tasks with a single output, the Jacobian is a vector and so does not contain pairwise node information. The Hessian, on the other hand, is an $N\times N$ matrix which does encode this information. Specifically, its elements denote the sensitivity of the output to each pair of input node features. In other words, for node-level tasks we use the first-order Taylor approximation, and for graph-level, since the Jacobian is unsuitable, we use the second-order, the minimal-order approximation to obtain any pairwise information between nodes.
>
> >*“'...solving a task' is not clearly defined … on real-world datasets, the authors evaluate four models, but none achieve perfect performance.”*
>
> We thank the reviewer for raising this interesting point, which we will clarify in the final version. We informally define a task as ‘solved’ when validation loss of a model approaches zero; Fig. 4 demonstrates this empirically, showing that as loss decreases, the model range approaches the known range of a synthetic task. Theoretical results linking the accuracy of the estimated range with validation error are an important direction which we reserve for future work.
>
> The reviewer is correct that, in our real-world experiments, models do not achieve perfect performance, meaning that the resulting range measures are only approximations of the true range of the underlying task. In the case of real-world tasks, the range is not just an approximation of the task range, but is also the true range of a model trained on that task. It is not our intention that a single range score be used to draw conclusions about a task. Instead, we infer conclusions about tasks by an examination of both the range and performance across different models: if range and performance correlate positively, it suggests a more long-range task, whereas no correlation suggests a task may be less long-range. In this way, our measure can serve as a practical tool for evaluating tasks for which no existing models can achieve perfect performance (i.e. all non-trivial tasks).
>
> >*“It would be valuable to empirically analyze the range of heterophilous tasks…”*
>
> We thank the reviewer for this suggestion. While we agree that it is worth investigating the relationship between label heterophily and long-range dependency, we are not aware of work explicitly making this connection. That said, we follow the reviewer’s suggestion and report the range during training for Roman-Empire and Amazon-Ratings in Figs. A1-2 in the linked pdf, taking hyper-parameterizations from [1]. We find that GCN and GT (which perform local convolutional and attentional message passing respectively) converge at a range of ~1 hop or lower for both tasks, but GT performs better, suggesting that range is a less important factor for these tasks than how information is propagated through the graph. This is further supported by the lack of correlation observed between range and performance between GCN and GCN+VN, especially for Roman-Empire.
>
> >*“The proposed method is based on influence scores… I suggest the authors discuss how this concept aligns with related work in the broader literature”*
>
> The suggested reference is indeed an important prior work. [2] introduced the notion of influence function to study the importance of certain training points on a model’s prediction, from which [3] adapted influence functions to the context of graph learning to study and improve information propagation in GNNs. A discussion of the paper along these lines will be included in the final version. We thank the reviewer for this suggestion, and will make sure to appropriately connect our work to the relevant broader literature.
>
> ---
> We thank the reviewer again for their feedback. We are happy to address any further concerns, and would be grateful if the reviewer would consider raising their score in light of our responses and additional experiments.
>
> [1] Platonov et al. (2023) https://arxiv.org/abs/2302.11640
>
> [2] Koh et al. (2017) https://arxiv.org/abs/1703.04730
>
> [3] Xu et al. (2018) https://arxiv.org/abs/1806.03536

---

### Decision · Program_Chairs · 2025-05-01

**Decision:**

Accept (poster)

**Comment:**

This paper addresses the problem of evaluating long-range dependencies in Graph Neural Networks (GNNs) by introducing a formal metric to measure the range of tasks and models. The proposed metric overcomes limitations of existing empirical benchmarks, such as the Long Range Graph Benchmark (LRGB), with a better theoretical grounding. Using synthetic datasets and real-world tasks like PASCAL-VOC and peptides tasks, the work shows the effectiveness of the new metric in quantifying a model's ability to capture long-range dependencies. The work provides a framework for studying and improving long-range interactions in GNNs, offering a better understanding of long-range tasks in graph machine learning and paving the way for better evaluations and GNN improvements.

The reviewers were generally positive. I support acceptance.